# Enhanced streamflow prediction with SWAT using support vector regression for spatial calibration: A case study in the Illinois River watershed, U.S.

**Lifeng Yuan**[1], **Kenneth J. Forshay**[2]*

**1** National Research Council Resident Research Associate at the United States Environmental Protection Agency, Robert S. Kerr Environmental Research Center, Ada, Oklahoma, United States of America, **2** U.S. Environmental Protection Agency, Center for Environmental Solutions and Emergency Response, Robert S. Kerr Environmental Research Center, Ada, Oklahoma, United States of America

* Forshay.Ken@epa.gov

**Data Availability Statement:** The data is available at doi.org/10.23719/1520734.

**Funding:** This research was funded by the United States Environmental Protection Agency as part of

## Abstract

Accurate streamflow prediction plays a pivotal role in hydraulic project design, nonpoint source pollution estimation, and water resources planning and management. However, the highly non-linear relationship between rainfall and runoff makes prediction difficult with desirable accuracy. To improve the accuracy of monthly streamflow prediction, a seasonal Support Vector Regression (SVR) model coupled to the Soil and Water Assessment Tool (SWAT) model was developed for 13 subwatersheds in the Illinois River watershed (IRW), U.S. Terrain, precipitation, soil, land use and land cover, and monthly streamflow data were used to build the SWAT model. SWAT Streamflow output and the upstream drainage area were used as two input variables into SVR to build the hybrid SWAT-SVR model. The Calibration Uncertainty Procedure (SWAT-CUP) and Sequential Uncertainty Fitting-2 (SUFI-2) algorithms were applied to compare the model performance against SWAT-SVR. The spatial calibration and leave-one-out sampling methods were used to calibrate and validate the hybrid SWAT-SVR model. The results showed that the SWAT-SVR model had less deviation and better performance than SWAT-CUP simulations. SWAT-SVR predicted streamflow more accurately during the wet season than the dry season. The model worked well when it was applied to simulate medium flows with discharge between 5 $m^3$ $s^{-1}$ and 30 $m^3$ $s^{-1}$, and its applicable spatial scale fell between 500 to 3000 $km^2$. The overall performance of the model on yearly time series is "Satisfactory". This new SWAT-SVR model has not only the ability to capture intrinsic non-linear behaviors between rainfall and runoff while considering the mechanism of runoff generation but also can serve as a reliable regional tool for an ungauged or limited data watershed that has similar hydrologic characteristics with the IRW.

the Office of Research and Development, Safe and Sustainable Water Research Program. Ken Forshay is a Research Ecologist of the Environmental Protection Agency. Lifeng Yuan was a National Research Council, Senior Research Associate, resident at the U.S. EPA working with Dr. Kenneth Forshay.

**Competing interests:** The authors have declared that no competing interests exist.

## Introduction

Reliable prediction of monthly streamflow can provide crucial information, assisting with decision making for watershed managers, such as future flood and drought forecasting, water quality evaluation and water resources optimization [1, 2]. However, the rainfall-runoff relationship has highly complex and non-linear hydrological features because the transformation from rainfall to runoff is influenced by various natural and human factors including precipitation, terrain, soil, land use and land cover (LULC), evapotranspiration, and groundwater, which makes it difficult to simulate and estimate streamflow with desirable accuracy [3–5]. Numerous hydrologic models with varying degrees of complexity have been developed to expound the rainfall-runoff relationship and predict runoff [6]. Hydrologic models can be roughly categorized into three groups: conceptual model (or grey-box model), physically-based model (or white-box model), and data-driven model (or black-box model) [7, 8]. Conceptual models consider primary hydrological components (e.g. precipitation, snow accumulation and melt, soil moisture storage, river routing, and reservoirs) and are built based on observed data or empirical formulation between many hydrological variables [5]. Conceptual models are helpful to understand the critical physical processes in the hydrological cycle. Physically-based models primarily concern the mathematical description of numerous physical processes in the hydrologic cycle (e.g. various partial or differential equations of expressing the physical laws of mass, energy, and momentum conservations). Physically-based models facilitate the comprehension of hydrological mechanisms but require a considerable amount of spatiotemporal data and model parameters input [3]. Data-driven models include empirical-based statistic models (e.g. various regression formulas) and artificial-intelligent-based models (e.g. artificial neural network (ANN), support vector machine (SVM), and other machine learning methods). They possess powerful predictive ability which accurately captures computable relationships between the relevant input and output variables but neglect detailed characteristics and processes of watershed systems and simplify the nonlinear relationship of rainfall-runoff [7]. In practice, however, there is no clear boundary to divide a single model into the mentioned-above three groups since a hydrological model is often built on multiple methods to improve their applications.

The Soil and Water Assessment Tool (SWAT) is a conceptual, physically-based, and basin-scale hydrologic model and has been extensively applied worldwide [9, 10]. Like many other physically-based hydrologic models, SWAT requires a large amount of data and parameter inputs to run. However, some data are difficult to collect due to time or economic cost, as well as the values of many parameters can only be obtained by calibration [11]. However, the process of calibration is typically time-consuming [12] and complicated as it involves parameterization, the selection of optimization algorithms, and extensive iterative simulations to find optimal parameter combinations and appropriate value range [13]. This challenge is extreme in the cases where limited data exist for parameterization and calibration [14].

SVM is a data-driven machine learning model that has been widely applied to hydrologic prediction, such as short-term or long-term streamflow and sediment yield forecasting [4, 15–19], water quality prediction [20, 21], precipitation, temperature and evapotranspiration simulation [22, 23], and the process of parameterization [12]. The essential characteristic of the SVM method is its ability to efficiently and accurately predict the nonlinear relationship between input and output variables without considering their internal physical links. Furthermore, SVM based on the structural risk minimization principle has shown to be a superior ANN based on the empirical risk minimization principle in several hydrological prediction applications [3, 4, 12]. SVM uses the kernel function and the maximum margin algorithm to solve the nonlinear problem through projecting an input space to a feature space where the

nonlinear problem is converted into the linear problem. Additionally, SVM typically applies a grid search method [8] to conduct hyperparameters optimization. The value of SVM applications in streamflow prediction includes how to find the optimal parameter set, raise prediction ability in the test data while keeping high accuracy in training data, as well as avoiding overfitting and uncertainty issues.

Although different categories of hydrologic models exist, streamflow prediction in an ungauged or watershed with limited monitoring data is still a challenging task in hydrology. SWAT can be applied to predict streamflow in an ungauged watershed, but the results of its application are hard to verify due to the lack of on-site data [24] and often there are underlying drivers of variability that are not contained in the typical calibration of a physical model [13]. Similarly, SVM and other machine learning will not perform well in streamflow simulation without a large amount of training data. If there are adjacent gauged watersheds (proxy watersheds) that have similar hydrological characteristics to those within an ungauged watershed, then we can use these proxy watersheds as donor watersheds and treat the ungauged watershed as the target watershed, and then conduct hydrological parameters transferability research [24–26]. The basis of this hypothesis is an application of the first law of geography [27] in which the climate and watershed conditions change smoothly over space and parameters in nearby regions. By integrating a hybrid SWAT-SVR approach, we can better capture the underlying variability of non-linear drivers while considering the hydrological processes. Hence, it is possible to build a SWAT-SVR hydrological model to predict streamflow in an ungauged target watershed with comparable proxy watersheds. In this article, we hypothesized that the application of SVM coupled to the physically-based SWAT model could help improve the model performance. We tested this hypothesis by comparing a common calibration approach SWAT Calibration and Uncertainty Programs (SWAT-CUP) with Sequential Uncertainty Fitting version 2 (SUFI2) algorithm to our hybrid SWAT-SVR method to develop models of streamflow at monthly time scales in the Illinois River watershed (IRW), USA.

Several works have evaluated the performance of SWAT and SVM in streamflow prediction [12, 19, 28]. Zhang, Srinivasan [12] et al. applied Artificial Neutral Network (ANN) and SVM methods to identify the optimal SWAT parameters to save the time cost of calibration and improve the efficiency of parameter calibration in two watersheds of the U.S. Jajarmizadeh, Kakaei Lafdani [28] et al. compared the monthly streamflow predictions from SWAT and SVM, and found the SVM model had a closer value for the average flow in comparison to the SWAT model. These efforts, however, either applied SVM in searching the optimal calibration parameters or built separate SWAT and SVR models, then estimated their running performance. Few studies have combined the two methods for a hybrid approach to streamflow prediction. Chiogna, Marcolini [19] et al. developed an SVM with SWAT model to predict hydropeaking in alpine watersheds in the Northeastern Italian. They used SVM to train the output of SWAT and found the SVM model can capture the fluctuation in streamflow. To the best of author's knowledge, no study has coupled the SVM and SWAT for streamflow prediction while considering wet-dry change. The objective of this study is to show how a support vector regression (SVR) method to support SWAT calibration can be used to improve monthly streamflow prediction for different seasons in the IRW.

## Materials and methods

### Study area

The IRW (35°31'-36°9'N, 94°12'-95°2'W) crosses Arkansas and Oklahoma, USA, separated almost equally by a state border, and has a drainage area of 4200 km². The basin elevation ranges from 121 to 602 meters above mean sea level. The average slope of the IRW is 5.6%, and

the slope ranges from 0 to 52.6%. The length of the Illinois River is approximately 230 km, flowing from Arkansas to Oklahoma before entering into Tenkiller Ferry Lake in Oklahoma [29]. Other large tributaries within the IRW include the Baron Fork Creek and the Flick Creek. The main soil types are Clarksville (43.8%), Rueter (26.9%) and Enders (18.6%) according to Soil Survey Geographic Database (SURRGO). The IRW is dominated by deciduous forest (40.7%) and pasture/hay (40.3%) as reported by the 2011 National Land Cover Dataset (NLCD).

The climate is humid in this region with an average annual temperature about 16˚C. The average yearly precipitation is 1198 mm. The mean annual lake evaporation is about 1270 mm [30]. Thirteen U.S. Geological Survey (USGS) hydrologic stations were selected to develop this new method. These monthly discharge data can be accessed and downloaded by USGS official website (https://dashboard.waterdata.usgs.gov/app/nwd/?region=lower48). Daily weather data of five climate stations from the National Climatic Data Center (NCDC) were used as weather input of the SWAT model. Fig 1 shows the spatial distribution of terrain, rivers, hydrologic and meteorological stations, and lakes in this area, and the relative position of the IRW in the U.S.

Although some studies have focused on the IRW [29, 31–34], these efforts paid more attention on water quality and nonpoint source pollution (NSP) evaluation, and few attempted to improve the accuracy of streamflow prediction. However, accurate streamflow simulation is a fundamental base for subsequent water quality and NSP simulation. In this study, we concentrated on improving the accuracy of streamflow at a monthly time scale through integrating a physically-based SWAT model and a data-driven SVR method.

## The SWAT model

SWAT is a continuous, semi-distributed, and physically-based hydrologic model used to simulate water cycles, crop growth, sediment yields, and agricultural chemical transport in a large river basin with varying soils, slopes and land use management conditions [9]. SWAT was developed by the U.S. Department of Agriculture Agricultural Research Service (USDA-ARS), and has been extensively used worldwide [10, 35]. In SWAT, a watershed is initially delineated into multiple sub-watersheds, then a sub-watershed is further divided into one or more hydrological response units (HRUs) where all land areas have similar land use, soil property, and slope combinations [36]. An HRU is the smallest spatial response unit where many physical processes such as hydrological cycle, soil erosion, nutrient and pesticide transport are simulated [37]. Primary input data include digital elevation model (DEM), land use, soil, and weather (i.e. precipitation, temperature, wind speed, solar radiation, and relative humidity). Water, sediment, and chemical movement in SWAT involve two phases: first, the watershed land areas control water transported to the channels together with sediment, nutrients and pesticides in each sub-watershed. Then, the movement of water and other mass through the stream network to the watershed outlet [38]. A more detailed description of the SWAT model can be available from online documentation (https://swat.tamu.edu/docs/).

## SWAT model setup

We used ArcSWAT version 2012.10_4.19 within ArcGIS 10.4.1 to build the IRW SWAT model. Digital elevation model (DEM) was obtained from Shuttle Radar Topography Mission (SRTM) 1 Arc-Second (about 30 m × 30 m) Global Database and downloaded from USGS website (https://earthexplorer.usgs.gov/, 01-28-2018) (Fig 2a). Land use and land cover (LULC) data was from the 2011 NLCD dataset (https://www.mrlc.gov/, 01-31-2018) (Fig 2b), and spatial resolution is 100 m × 100 m. Soil data came from the SSURGO database (https://websoilsurvey.nrcs.usda.gov/, 02-05-2018) (Fig 2c). Climate data obtained from the

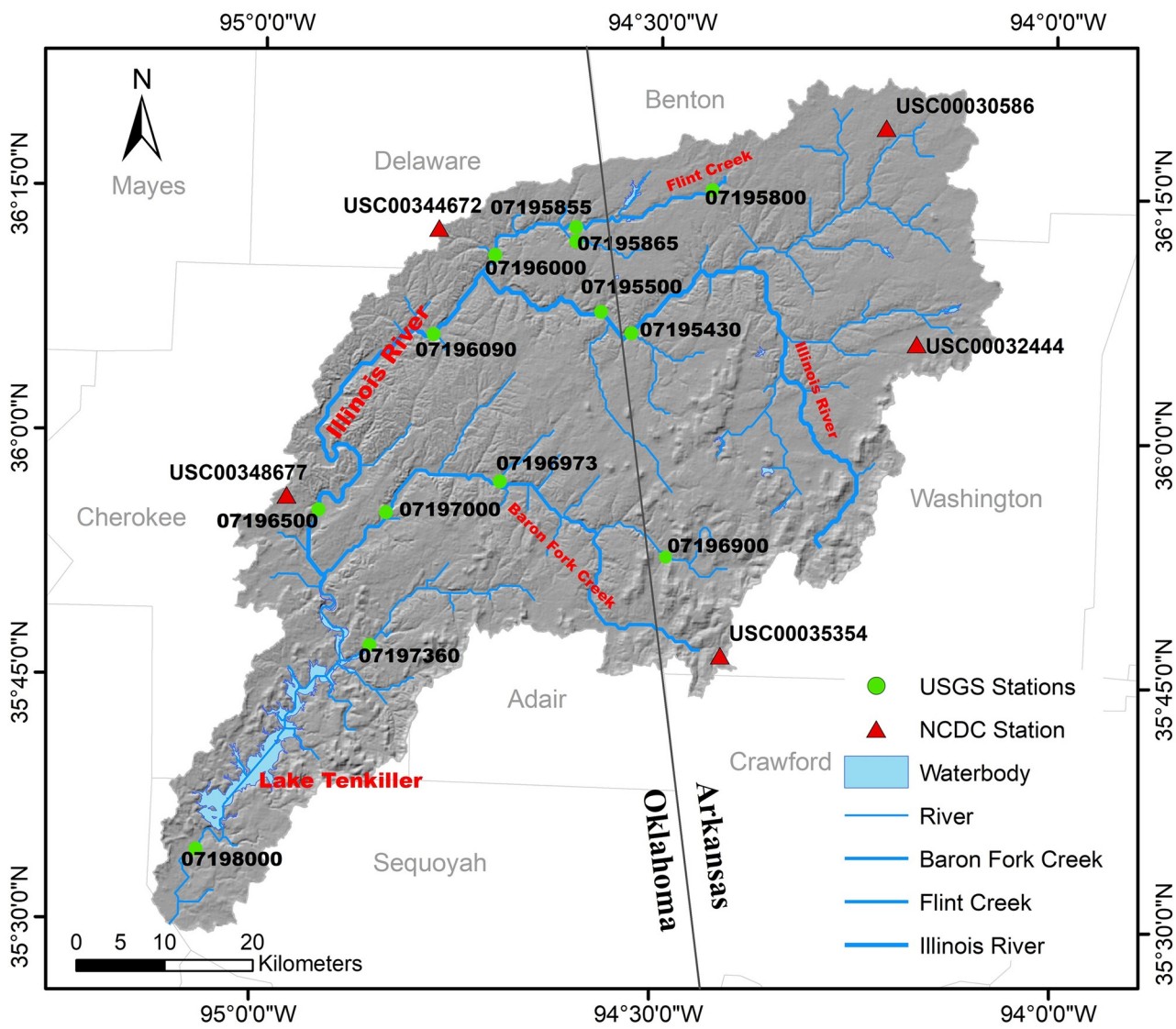

**Fig 1. The geographic position of USGS hydrological stations, NCDC meteorological stations, and principal rivers in the IRW.**

National Climatic Data Center (NCDC) (https://www.ncdc.noaa.gov/, 02-07-2018) (Fig 2d). Due to missing precipitation and temperature records from NCDC climate data from Jan. 1990 to Dec. 2013, we downloaded alternative Climate Forecast System Reanalysis (CFSR) data from the SWAT official website (https://globalweather.tamu.edu/, 01-31-2018), then filled missing NCDC data using climate data from the closest CFSR grid stations (not shown in Fig 2d). All precipitation data of five climate stations meet the data consistency checks using the double mass curve method [39]. The basic information of thirteen hydrologic stations is listed in Table 1.

The IRW was delineated into 86 subwatersheds with 1023 HRUs under a threshold area of 3000 ha. The multiple land use/soil/slope method was applied to define the HRUs with land use (10%), soil (10%) and slope (5%) threshold. The surface runoff was estimated using the SCS curve number method [40], and the Penman-Monteith equation [41] was applied to calculate the potential evapotranspiration. The streamflow was routed and calculated according

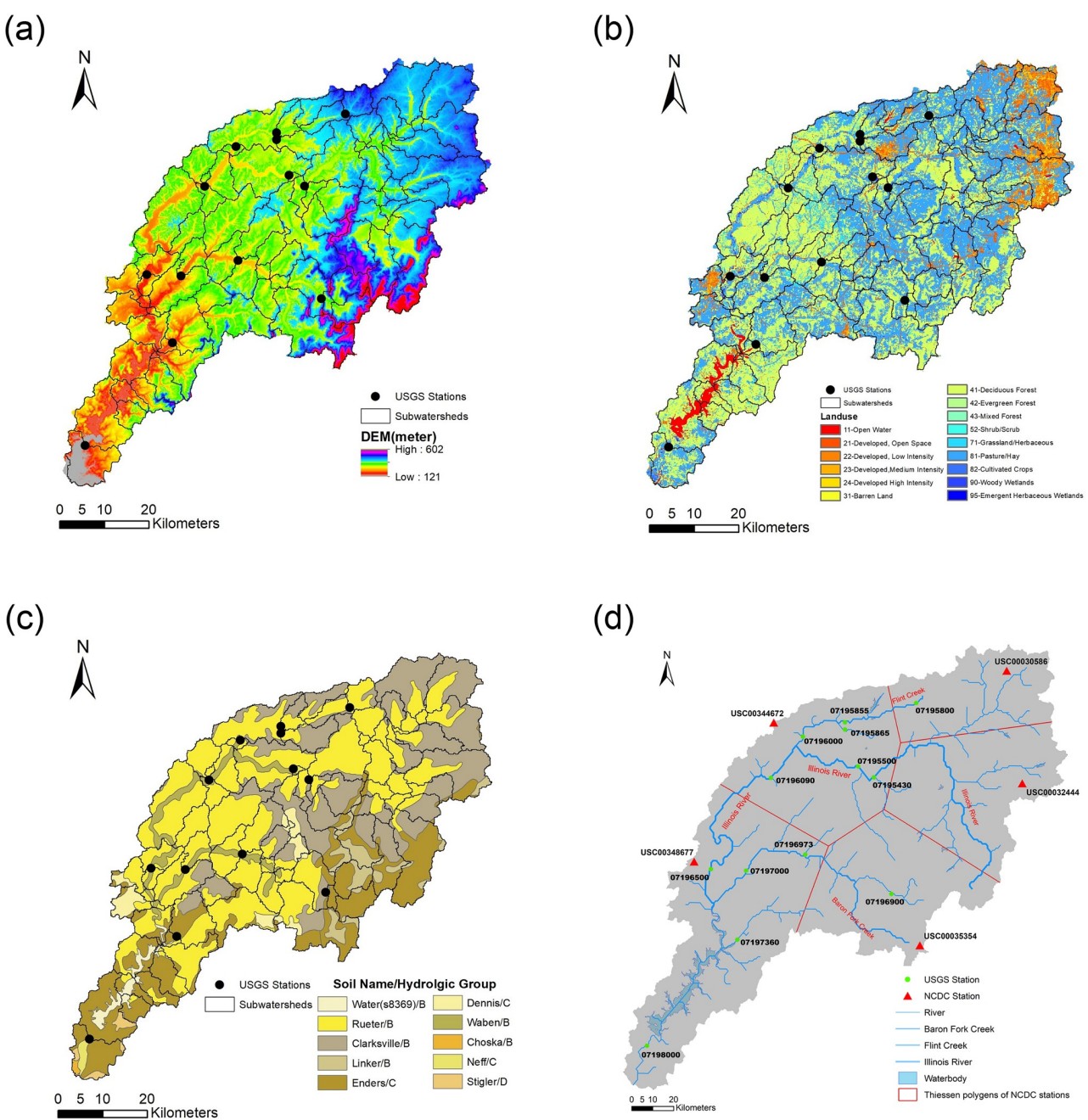

**Fig 2. a) DEM, b) LULC, c) Soil, and d) USGS station, river and weather station map in the IRW.**

to the variable storage routing method [38]. A five-year was used as a warm-up period (1990–1994) to initialize the model input and stabilize the SWAT model. The simulation running period of the SWAT model is from Jan-01-1995 to Dec-31-2013.

## Streamflow prediction

**Dividing dry and wet season.** There is evidence that SWAT model performance can be improved and better reflect the seasonal change of parameters by separating the dry and wet

**Table 1. Watershed properties of selected USGS stations.**

| No. | USGS station (Subwatershed No.) | Upstream area (km²) | Simulated upstream area† (km²) | Data period | Number of data | Average monthly streamflow (m³ s⁻¹) | Group |
|---|---|---|---|---|---|---|---|
| 1 | 07195800 (1) | 36.8 | 36.2 | 1.1995–12.2013 | 228 | 0.42 | Low flows |
| 2 | 07195855 (7) | 155.0 | 134.5 | 1.1995–12.2013 | 228 | 1.27 | Low flows |
| 3 | 07195865 (12) | 49.5 | 52.8 | 1.1997–12.2013 | 204 | 0.68 | Low flows |
| 4 | 07196000 (17) | 300.7 | 302.8 | 1.1995–12.2013 | 228 | 3.01 | Low flows |
| 5 | 07195500 (24) | 1633.0 | 1570.2 | 1.1995–12.2013 | 228 | 18.71 | Medium flows |
| 6 | 07195430 (26) | 1490.5 | 1438.0 | 1.1996–12.2013 | 216 | 17.68 | Medium flows |
| 7 | 07196090 (28) | 2138.5 | 2072.8 | 7.2010–12.2013 | 42 | 25.47 | Medium flows |
| 8 | 07196973 (46) | 64.8 | 66.0 | 1.1995–12.2002 | 96 | 0.73 | Low flows |
| 9 | 07196500 (51) | 2462.5 | 2385.8 | 1.1995–12.2013 | 228 | 27.76 | Medium flows |
| 10 | 07197000 (52) | 808.7 | 797.1 | 1.1995–12.2013 | 228 | 9.21 | Medium flows |
| 11 | 07196900 (62) | 105.2 | 105.2 | 1.1995–12.2013 | 228 | 1.31 | Low flows |
| 12 | 07197360 (74) | 233.8 | 228.3 | 1.1998–12.2013 | 192 | 2.41 | Low flows |
| 13 | 07198000 (85) | 4186.2 | 4070.0 | 1.1995–12.2013 | 228 | 44.03 | High flows |

†Note: The column of the simulated upstream area refers to delineate the upstream area by the ArcSWAT program.

seasons [42–44]. Therefore, we developed the SWAT-SVR model based on the separation of the dry and wet seasons to reflect the impact of seasonal change. In this paper, we used the run-off coefficient (RC) of subwatersheds and flow discharge at the outlet of subwatersheds to divide the dry and wet seasons. The RC is calculated by dividing the areally averaged total monthly runoff by the areally averaged total monthly rainfall. The areally averaged total monthly runoff is computed by multiplying flow rate measured at the watershed outlet with time then dividing by the watershed area. The Thiessen polygons of NCDC stations in Fig 2d were used to partition the IRW. Daily rainfall from NCDC stations was aggregated by month to represent the areally averaged total monthly rainfall in each Thiessen polygon region. The statistic period of data at each station can be found in Table 1.

Fig 3 Shows the distinction between wet and dry seasons of rainfall-runoff characteristics. The average monthly RC (AMRC) of the IRW was 0.3. The maximum and minimum AMRC were 0.45 and 0.11, which occurred in January and September. The AMRC before and after June was 0.39 (purple line in Fig 3) and 0.2 (green line in Fig 3). The AMRC gradually declines from January to September, then quickly increases afterward (red line in Fig 3). January to April were the months of the highest AMRC, and August to October were the months of the lowest AMRC. The AMRC at the subwatershed 28 did not follow the common trend of most subwatersheds because the data length of 07196090 site only came from 42 months, and it is far less than the other twelve sites.

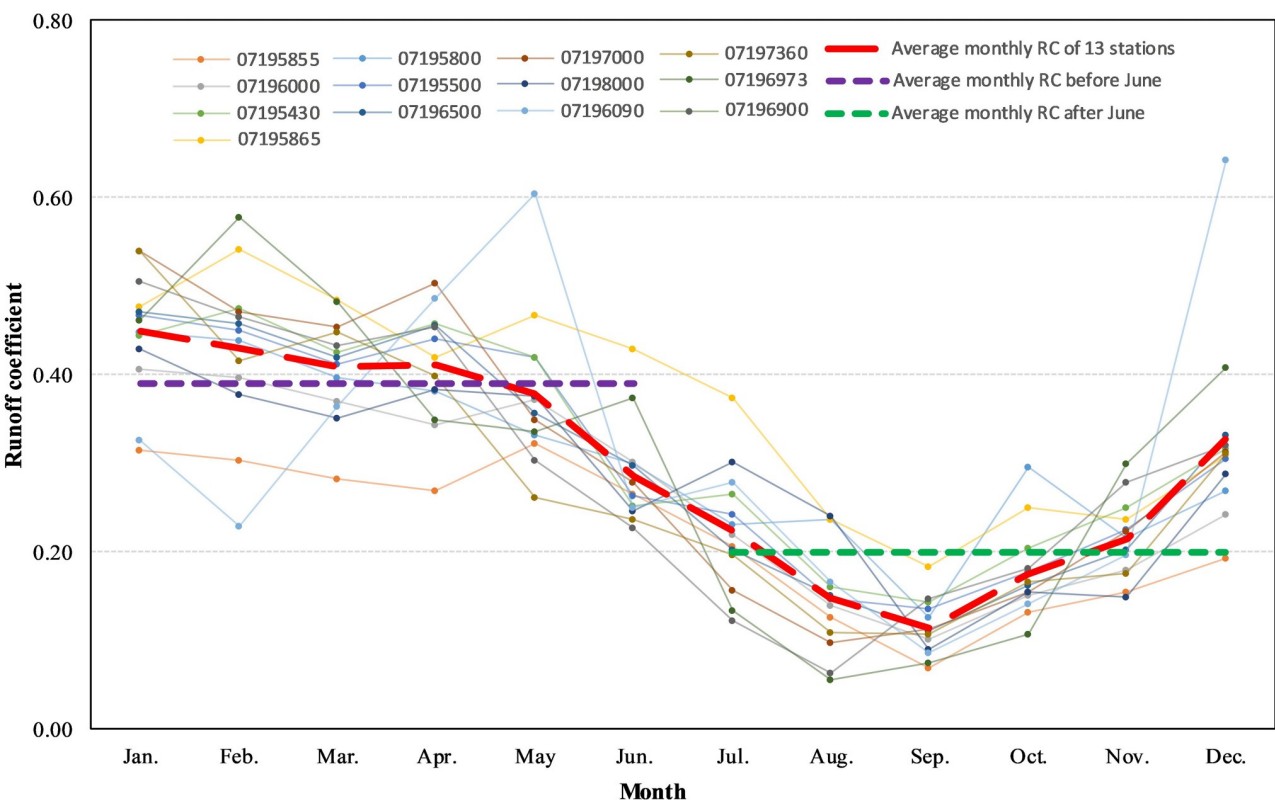

**Fig 3. Average monthly runoff coefficient on 13 subwatersheds in the IRW.**

To illustrate the distribution of monthly streamflow on thirteen stations, we plotted the average monthly streamflow hydrograph (Fig 4). Streamflow was categorized into three groups based on the volume of flows discharge: low flows, medium flows, and high flows (Table 1). Low flows with discharge less than 5 m$^3$ s$^{-1}$ come from 07195800, 07195855, 07195865,

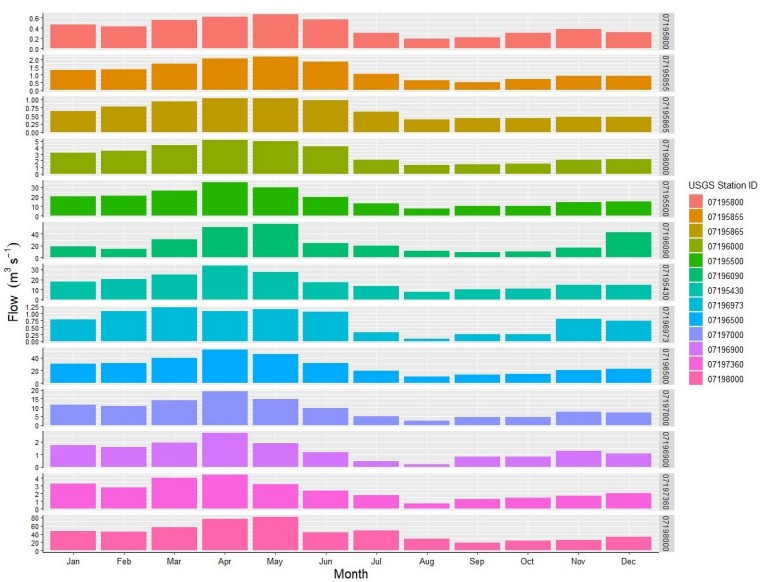

**Fig 4. Average monthly streamflow at 13 hydrologic stations.**

07196000, 07196973, 07196900, and 07197360; Medium flows with discharge between 5 m$^3$ s$^{-1}$ and 30 m$^3$ s$^{-1}$ are from 07195500, 07196090, 07195430, 07196500, and 07197000; High flows with discharge greater than 30 m$^3$ s$^{-1}$ are from 07198000. Average monthly maximum and minimum streamflow occurred at stations 07198000 (the outlet of the IRW) and 07195800 (the subwatershed of the most upper reach), respectively. The maximum and minimum discharge occurred in April and September (Fig 4). Streamflow from January to June accounted for 67.39% of the annual total amount, which is approximately two times greater than those from July to December. Based on the analysis of RC and flows, we divided January to June as the wet season and July to December as the dry season.

**Coupling SWAT with SVR.** To improve monthly streamflow prediction, we combined the SWAT model and SVR method and developed the SWAT-SVR model. In this approach, the outcome of flow (including baseflow) was first simulated by SWAT with its default parameter combinations without the calibration procedure. Then, the simulated streamflow at month *t* from the SWAT model and the upstream drainage area of the station serve as two inputs of the SVR model to predict streamflow on month *t*. This design reduced time needed to calibrate and validate the SWAT model as well as time of features selection during the SVR application. In this design, SWAT was regarded as a comprehensive transfer function by integrating weather, terrain, LULC, soil data, and producing new flow output that serves as input to the SVR model.

SVM is a black box, mathematic model, which attempts to search for an optimal separating hyperplane with the maximal margin between observations and finds the optimal function and parameter sets fitting the observations while avoiding overfitting and having better generalization ability [19]. SVR belongs to an application of SVM for regression analysis. A detailed description of SVM theory is beyond the scope of this article, and it can be obtained from Vapnik [45], Hastie, Tibshirani [46], Chang and Lin [47], and Smola and Schölkopf [48].

The principle of SVM is rooted in the statistical learning and structural risk minimization theory [45]. Briefly SVM coverts a complex nonlinear problem in the original input space (i.e. the space of the observed data) into a simple linear problem in the feature space (i.e. some higher dimensional space) using a kernel function [49]. Commonly used kernel functions include the linear, polynomial, Gaussian radial basis (RBF), and sigmoid. Among these kernels, the linear kernel is a particular case of RBF, the sigmoid kernel behaves like RBF for certain parameters, and the polynomial kernel will produce more hyperparameters than the RBF kernel which causes more computational difficulties [50, 51]. Hence, we chose the Gaussian RBF kernel function, and its mathematic expression is described as:

$$K(x_i, x_j) = \exp(-\gamma \|x_i - x_j\|^2) \tag{1}$$

In an SVR ε-regression application based on RBF kernel, three parameters need to be determined: the penalty parameter of the error term C (C > 0), the Gaussian RBF kernel parameter γ, and the width/deviation of the error margin ε. The grid search and the k-fold cross-validation method were used to optimize these parameters by defining the upper and lower bound for each parameter and estimating the predicted accuracy of the model. In the *k*-fold cross-validation, the dataset was subdivided into *k* subsets of nearly equal size. In each step, the *k*-1 subsets were used to train the model while the remaining subset was used for validation [19]. Each subset was applied exactly once for validation. At last, the averaged error of all *k* trials was calculated. In our study, we first chose a coarse numeric range of C, γ, and ε to conduct grid search, then narrowed down this search range according to the output of the SVR model. R version 3.4.0 running on RStudio version 1.1.456 and the 'e1071' package [52] were used for the development, training and testing of the SWAT-SVR model. Standardizing data can avoid

numbers in greater ranges dominating those in smaller ranges and reduce calculation complexity [51]. Also, a scaling tool in the 'e1071' package does not work very well for SVR regression analysis. Before building the model, hence, we normalized two input variables (i.e. streamflow and upstream drainage area) using Eq 2.

$$x_{new,i} = \frac{x_i - x_{min}}{x_{max} - x_{min}} \tag{2}$$

where $x_{new,i}$ is the normalized parameter, $x_i$ is observed data series, and $x_{max}$ and $x_{min}$ are the maximum and minimum of the observation. Independent seasonal SWAT-SVR models were developed for monthly flow prediction at 13 stations. In each model run, the leave-one-out sampling method was applied to calibrate the SWAT-SVR model spatially. Out of $n$ stations, one station was excluded for testing purposes, and the SWAT-SVR model was trained with the remaining ($n$-1) stations. This step was repeated until all stations had been removed once [24].

**SWAT-CUP.** SWAT-CUP, a standalone SWAT calibration procedure [13], was used to compare the results of SWAT-SVR streamflow prediction. Parameters sensitivity analysis was conducted by the all-at-a-time approach with 1000 SWAT-CUP simulations. SUFI2 was employed into sensitivity analysis, calibration and validation to seek an optimized parameter set due to the high effectiveness of this algorithm [53]. SWAT and SWAT-CUP were run for all stations at one time with three iterations during the wet and dry seasons. After the first two iterations with 250 simulations for each iteration, parameter ranges were narrowed down by considering both the physical limitations of parameters and suggested ranges from the calibration. We applied the calibrated parameter ranges, and independent data from the station left out to conduct another iteration with 250 simulations for validation. The procedures of calibration and validation followed the guidelines of Moriasi et al. [54]. Fig 5 demonstrates a research flowchart describing the methodology used in this study.

## Model performance evaluation

We used $R^2$ (Pearson's coefficient of determination), NSE (Nash-Sutcliffe efficiency), PBIAS (percent bias), RMSE (root mean square error), and RSR (RMSE-observation's standard deviation ratio) to evaluate the model performance. $R^2$ and NSE are widely used as a reliable criterion to evaluate the predictive ability of hydrological models [55]. PBIAS measures the average magnitude of the simulations to be larger or smaller than their observations. In this study, positive values of PBIAS indicate the overestimation bias, and negative values refer to the underestimation bias. RMSE shows the discrepancy between the observed and simulated series. RSR indicates the residual variation of the prediction [56]. The lower RSR, PBIAS, and RMSE, the higher $R^2$ and NSE, and the better the model prediction performance. The 'hydroGOF' package in R was used to calculate the mentioned statistical indicators [57]. Table 2 listed the evaluation indicators and their calculation methods.

In this work, we applied a rating metric of hydrologic model evaluation from Moriasi et al. [54] to estimate the model performance (Table 3).

## Results and discussion

### Performance comparison between SWAT-SVR and SWAT-CUP

A total of 52 independent SWAT-SVR models were developed for monthly streamflow prediction (i.e. calibration and validation) during the wet and dry seasons in 13 USGS hydrologic stations. The corresponding 52 simulation results from SWAT-CUP were regarded as comparison experiments estimating the model performance. Spatial calibration method was implemented for each site. In each run, streamflow time series data from 12 stations were treated as

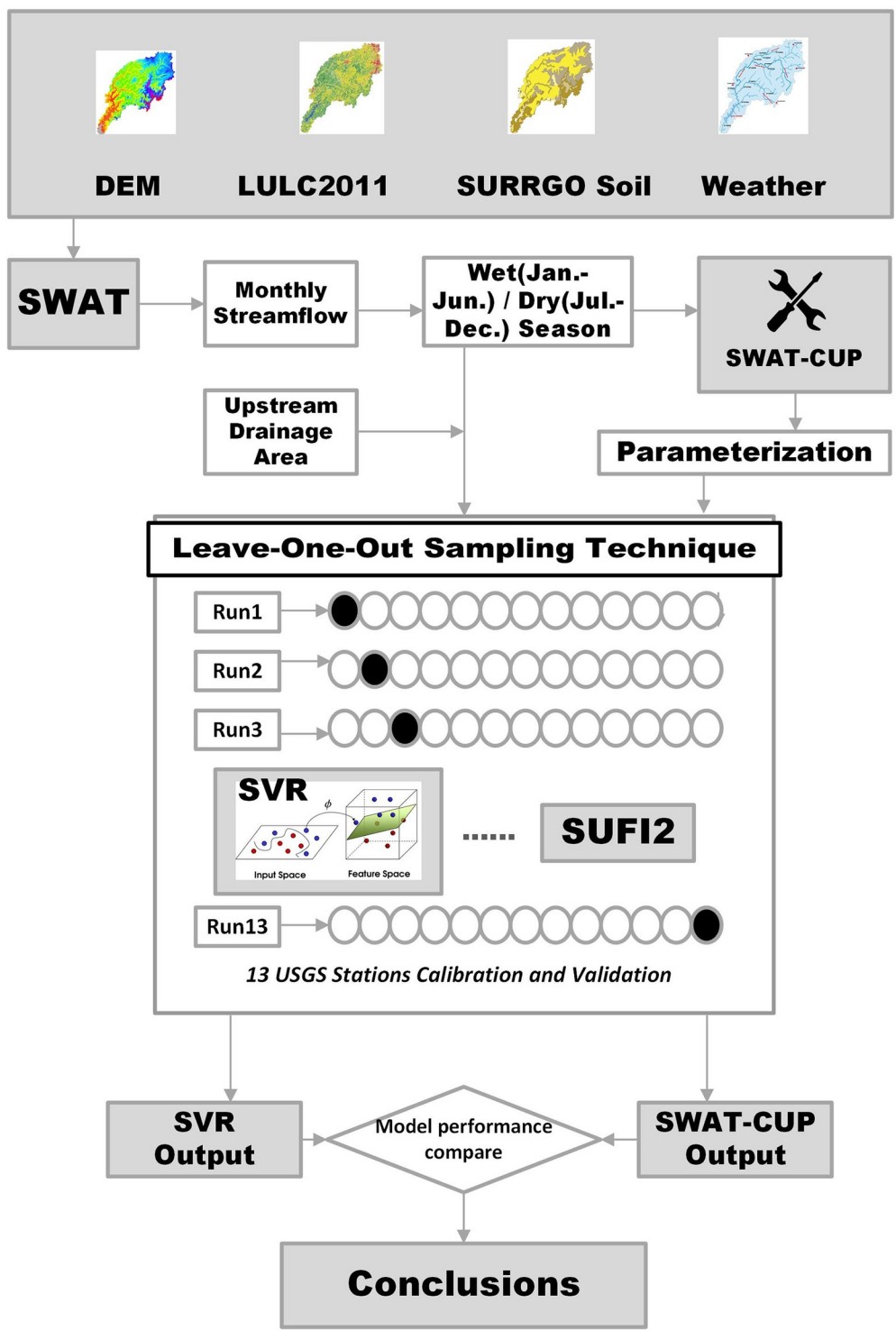

**Fig 5. Research flowchart of streamflow prediction by SWAT-SVR and SWAT-CUP.**

**Table 2. Evaluation indicators of the model performance and their mathematic expressions†.**

| Indicator Name | Calculation Equation | Description |
|---|---|---|
| **Pearson's coefficient of determination (R²)** | $R^2 = \frac{(\sum_{i=1}^{n}(y_i - \bar{y})(y_i' - \bar{y'}))^2}{\sum_{i=1}^{n}(y_i - \bar{y})^2 \sum_{i=1}^{n}(y_i' - \bar{y'})^2}$ | Range [0,1], and 1 is the perfect value (p.v.) |
| **Nash-Sutcliffe efficiency (NSE)** | $NSE = 1 - \frac{\sum_{i=1}^{n}(y_i - y_i')^2}{\sum_{i=1}^{n}(y_i - \bar{y})^2}$ | Range (-∞,1], and 1 is the p.v. |
| **Percent Bias (PBIAS)** | $PBIAS = 100 \times \frac{\sum_{i=1}^{n}(y_i' - y_i)}{\sum_{i=1}^{n} y_i}$ | Range (-∞, +∞), and 0 is the p.v. |
| **RMSE-observations standard deviation ratio (RSR)** | $RSR = \frac{\sqrt{\sum_{i=1}^{n}(y_i - y_i')^2}}{\sqrt{\sum_{i=1}^{n}(y_i - \bar{y})^2}}$ | Range [0, +∞), and 0 is the p.v. |
| **Root Mean Square Error (RMSE)** | $RMSE = \sqrt{\frac{\sum_{i=1}^{n}(y_i - y_i')^2}{n}}$ | Range [0, +∞), and 0 is the p.v. |

†Note: $y_i$ is the observed data series, $y_i'$ is the simulated results series, the overbar represents the mean value of data series, and n is the sample number.

training data, and the station left out was used for testing purpose. The support vector ε-regression based on the Gaussian RBF kernel was applied for developing the SWAT-SVR model. The initial numeric range of parameters in SVR grid searching is: C (begin = $2^{-6}$, end = $2^8$, step = 1), $\gamma$ (begin = $2^4$, end = $2^{-8}$, step = -1), and ε (begin = $2^{-8}$, end = $2^{-1}$, step = 0.5). The following fine search used a smaller step and range for the above three parameters according to variable results on different models. The final value range of C for 13 SWAT-SVR models are from 53.0156 to 255.0156, the value of $\gamma$ is 0.4, and ε is 0.00390625 for the wet season; the value of C falls between 32.0156 and 255.0156, $\gamma$ is 1.2, and ε is 0.00390625 as well in the dry season. The $k$-value in cross-validation was set 5 for SVR simulations.

Table 4 shows the calibration results of the model by SWAT-SVR and SWAT-CUP methods during the wet and dry seasons. According to Moriasi et al. [54], we conducted rigid criteria for the evaluation of the model performance (i.e. the overall performance of the model should be determined conservatively as the lowest rating when the value of RSR, NSE, and PBIAS has a conflicting performance). Table 4 indicates that 100% (13/13) of the SWAT-SVR runs for the wet season and 84.6% (11/13) of the runs for the dry season had "Good" performance ratings in calibration. Based on the value of PBIAS, the SWAT-SVR model slightly underestimated monthly streamflow for each watershed during the wet and dry seasons, and SWAT-CUP method also underestimated wet season streamflow but remarkably overestimated dry season streamflow. The mean of NSE and $R^2$ of 13 stations decreased from 0.92 and 0.92 in the wet season to -0.16 and 0.55 in the dry season, respectively. This results are consistent with Zhang, Chen [43]'s study in which the SWAT model can produce good simulations for the wet season but poor simulations for the dry season. The possible reason is that $R^2$ and NSE are sensitive to extremely large number (i.e. high flows took place in the wet season). Compared with the performance of SWAT-CUP, the SWAT-SVR model has approximately similar performances for the wet and dry seasons. We noted that the variation of statistics is

**Table 3. Performance ratings of recommended statistics for streamflow simulations e.g Moriasi et al [54].**

| Performance Rating | RSR | NSE | PBIAS (%) |
|---|---|---|---|
| **Very Good** | $0 \le RSR \le 0.5$ | $0.75 < NSE \le 1$ | $PBIAS < \pm 10$ |
| **Good** | $0.5 < RSR \le 0.6$ | $0.65 < NSE \le 0.75$ | $\pm 10 \le PBIAS < \pm 15$ |
| **Satisfactory** | $0.6 < RSR \le 0.7$ | $0.5 < NSE \le 0.65$ | $\pm 15 \le PBIAS < \pm 25$ |
| **Unsatisfactory** | $RSR > 0.7$ | $NSE \le 0.5$ | $PBIAS \ge \pm 25$ |

**Table 4. Calibration performance of streamflow simulations by SWAT-SVR and SWAT-CUP during the wet and dry seasons.**

| Station | | SWAT-SVR | | | | | | SWAT-CUP | | | | | |
|---|---|---|---|---|---|---|---|---|---|---|---|---|---|
| | | *RSR* | *NSE* | *PBIAS* | *R²* | *RMSE* (m³ s⁻¹) | Rating | *RSR* | *NSE* | *PBIAS* | *R²* | *RMSE* (m³ s⁻¹) | Rating |
| Wet season | 07195800 | 0.49 | 0.76 | -11.0 | 0.77 | 13.95 | Good | 0.39 | 0.85 | -12.5 | 0.85 | 11.80 | Good |
| | 07195855 | 0.50 | 0.75 | -11.6 | 0.76 | 14.19 | Good | 0.39 | 0.85 | -12.6 | 0.85 | 11.83 | Good |
| | 07195865 | 0.49 | 0.76 | -10.6 | 0.76 | 13.91 | Good | 0.25 | 0.94 | -7.2 | 0.94 | 7.62 | Very Good |
| | 07196000 | 0.49 | 0.76 | -10.1 | 0.77 | 13.92 | Good | 0.25 | 0.94 | -7.0 | 0.94 | 7.55 | Very Good |
| | 07195500 | 0.49 | 0.76 | -10.8 | 0.77 | 13.48 | Good | 0.29 | 0.92 | -11.8 | 0.92 | 8.49 | Good |
| | 07195430 | 0.49 | 0.76 | -11.3 | 0.77 | 13.42 | Good | 0.28 | 0.92 | -10.3 | 0.93 | 8.15 | Good |
| | 07196090 | 0.51 | 0.74 | -13.4 | 0.76 | 13.65 | Good | 0.18 | 0.97 | -5.1 | 0.97 | 5.08 | Very Good |
| | 07196973 | 0.51 | 0.74 | -13.2 | 0.76 | 14.08 | Good | 0.25 | 0.94 | -7.5 | 0.94 | 7.46 | Very Good |
| | 07196500 | 0.50 | 0.75 | -11.7 | 0.76 | 13.63 | Good | 0.27 | 0.93 | -8.8 | 0.93 | 7.43 | Very Good |
| | 07197000 | 0.49 | 0.76 | -10.6 | 0.77 | 13.85 | Good | 0.28 | 0.92 | -11.0 | 0.92 | 8.61 | Good |
| | 07196900 | 0.50 | 0.75 | -11.5 | 0.76 | 14.14 | Good | 0.25 | 0.94 | -7.5 | 0.94 | 7.66 | Very Good |
| | 07197360 | 0.49 | 0.76 | -11.2 | 0.76 | 13.96 | Good | 0.25 | 0.94 | -7.6 | 0.94 | 7.60 | Very Good |
| | 07198000 | 0.42 | 0.83 | -12.0 | 0.84 | 8.84 | Good | 0.25 | 0.94 | -12.6 | 0.94 | 5.39 | Good |
| Dry season | 07195800 | 0.55 | 0.69 | -12.3 | 0.70 | 8.57 | Good | 0.94 | 0.12 | 20.6 | 0.16 | 14.54 | Unsatisfactory |
| | 07195855 | 0.55 | 0.70 | -11.9 | 0.70 | 8.56 | Good | 1.02 | -0.03 | 94.4 | 0.63 | 15.78 | Unsatisfactory |
| | 07195865 | 0.55 | 0.69 | -12.5 | 0.70 | 8.54 | Good | 0.96 | 0.08 | 14.9 | 0.11 | 14.84 | Unsatisfactory |
| | 07196000 | 0.55 | 0.70 | -11.7 | 0.71 | 8.56 | Good | 1.01 | -0.03 | 94.1 | 0.63 | 15.76 | Unsatisfactory |
| | 07195500 | 0.55 | 0.69 | -12.1 | 0.70 | 8.39 | Good | 1.01 | -0.03 | 97.8 | 0.63 | 15.38 | Unsatisfactory |
| | 07195430 | 0.58 | 0.66 | -16.1 | 0.70 | 8.82 | Satisfactory | 1.02 | -0.05 | 102.2 | 0.64 | 15.50 | Unsatisfactory |
| | 07196090 | 0.55 | 0.70 | -12.6 | 0.71 | 8.24 | Good | 1.01 | -0.01 | 95.1 | 0.63 | 15.13 | Unsatisfactory |
| | 07196973 | 0.55 | 0.70 | -12.9 | 0.71 | 8.38 | Good | 1.18 | -0.40 | 124.1 | 0.63 | 17.95 | Unsatisfactory |
| | 07196500 | 0.59 | 0.65 | -16.6 | 0.69 | 8.51 | Satisfactory | 1.15 | -0.33 | 123.6 | 0.65 | 16.63 | Unsatisfactory |
| | 07197000 | 0.55 | 0.70 | -12.8 | 0.71 | 8.51 | Good | 1.19 | -0.41 | 122.2 | 0.61 | 18.38 | Unsatisfactory |
| | 07196900 | 0.55 | 0.69 | -12.5 | 0.70 | 8.58 | Good | 1.19 | -0.41 | 123.6 | 0.63 | 18.44 | Unsatisfactory |
| | 07197360 | 0.55 | 0.70 | -11.7 | 0.71 | 8.49 | Good | 1.18 | -0.40 | 123.6 | 0.63 | 18.28 | Unsatisfactory |
| | 07198000 | 0.40 | 0.84 | -12.2 | 0.85 | 3.98 | Good | 1.10 | -0.22 | 114.5 | 0.63 | 10.85 | Unsatisfactory |

small, and the value of each indicator is close between different SWAT-SVR models. This is because a single SWAT-SVR watershed model was built based on eleven other common watersheds in calibration. Although the simulation results from SWAT-CUP had better overall performance (i.e. 53.8% of the runs had "Very Good" ratings) than those of the SWAT-SVR model in the wet season, SWAT-CUP failed to estimate monthly streamflow in the dry season, in which all runs were identified as "Unsatisfactory" ratings. We are not surprised that the SWAT-SVR model has a good performance in the period of calibration because SVR typically possesses a strong learning ability for the training dataset. In the following section, we focus on the discussion of the model performance in validation and expect that the SWAT-SVR model has better generalization ability and can be applied in an ungauged watershed.

Fig 6 The performance ratings of the SWAT-SVR and SWAT-CUP model during wet and dry season validation. The values of NSE for 07196000 station from SWAT-SVR and SWAT-CUP are below zero in validated simulations. Hence, site 07196000 is not shown on the figures for clarity. The subsequent analysis only showed 12 valid stations. Fig 6a shows that 75% (9/12) of SWAT-SVR model prediction for the wet season falls into the ratings of "Good" and "Satisfactory", and the performance ratings of three models are "Unsatisfactory". In comparison with SWAT-SVR, 66.7% of SWAT-CUP simulations belong to "Good" and "Satisfactory", and the ratings of four models are "Unsatisfactory". Although 50% of all models had consistent

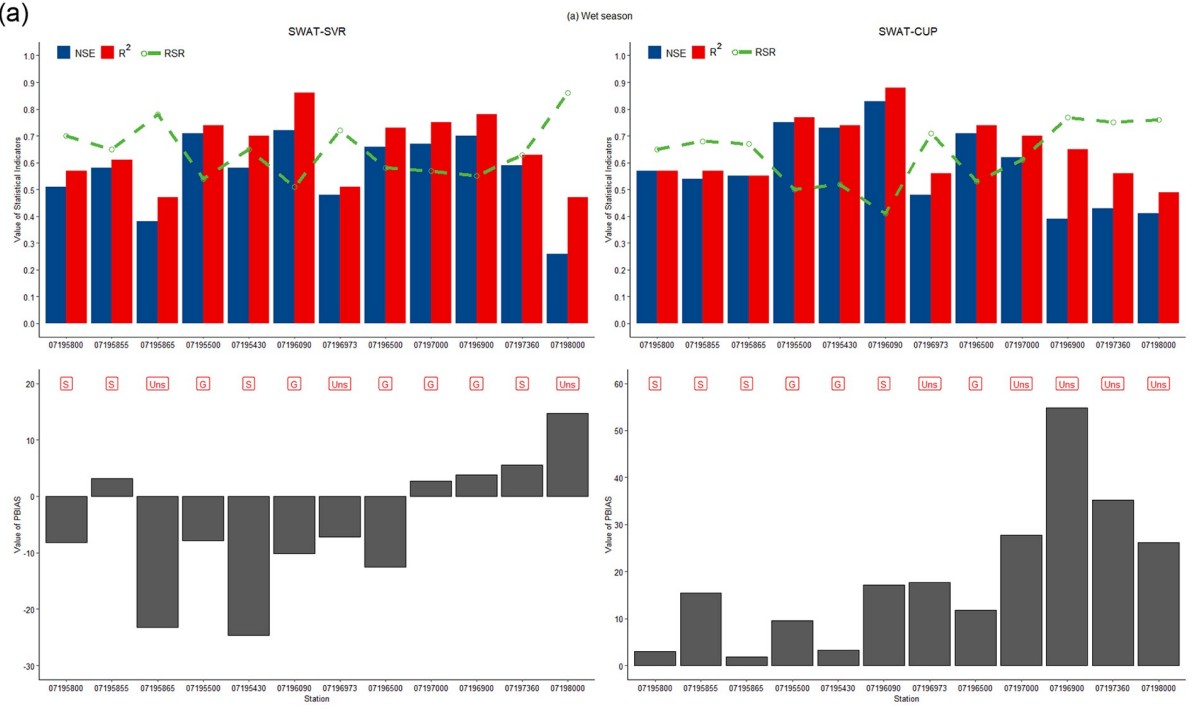

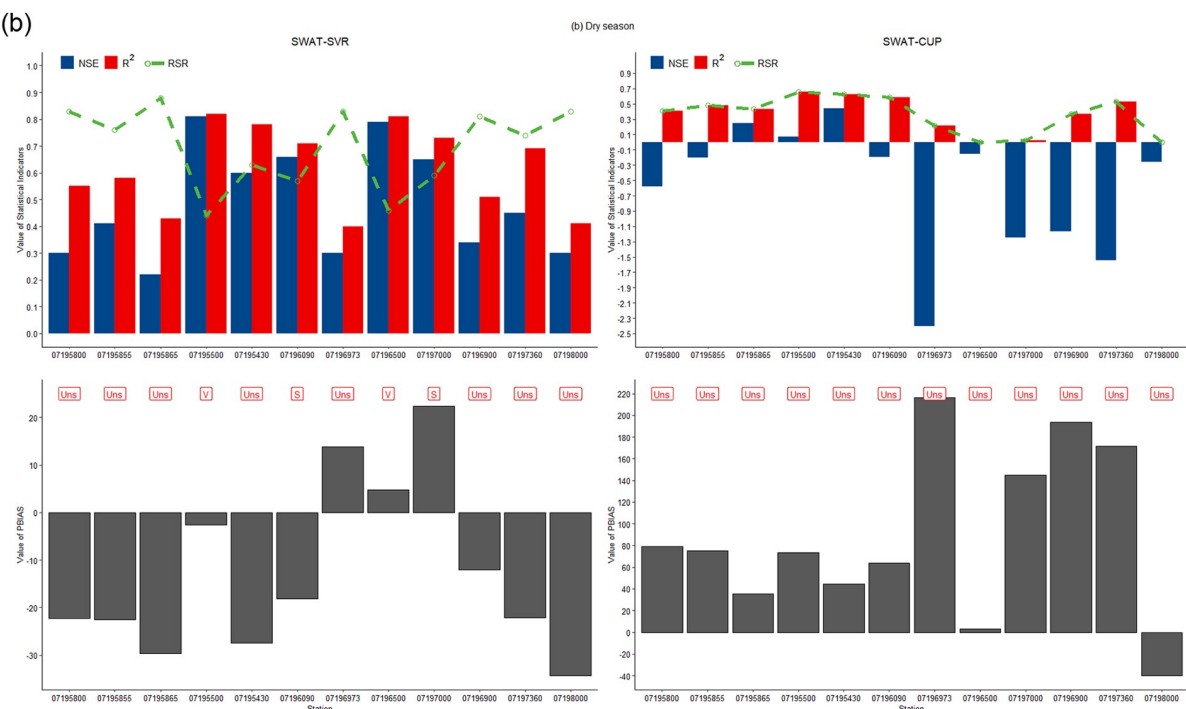

**Fig 6. Comparison of performance ratings during the wet (a) and dry (b) seasons between SWAT-SVR and SWAT-CUP.** (Performance Rating: V: Very good, G: Good, S: Satisfactory, Uns: Unsatisfactory).

ratings produced by SWAT-SVR and SWAT-CUP, more "Good" and less "Unsatisfactory" ratings were observed in the SWAT-SVR model. In the wet season, the average RSR, NSE, PBIAS, $R^2$, and RMSE from SWAT-SVR and SWAT-CUP is 0.65, 0.57, -5.33, 0.65, and 10.45 $m^3$ $s^{-1}$, and 0.63, 0.58, 18.63, 0.65, and 9.88 $m^3$ $s^{-1}$, respectively. We concluded that the SWAT-SVR model has less discrepancy (i.e. the smaller absolute value of PBIAS) than SWAT-CUP simulations despite very close values from other statistics, and SWAT-SVR slightly underestimated wet season streamflow.

In the dry season, the SWAT-SVR model had better model performance than SWAT-CUP simulations according to Fig 6b. Two SWAT-SVR models (07195500 and 07196500) had "Very Good" ratings, and the other two models (07196090 and 07197000) obtained "Satisfactory" ratings. No "Satisfactory" or better performance exists in SWAT-CUP simulations, and this result is coherent and consistent with the performance of SWAT-CUP in calibration. The average RSR, NSE, PBIAS, $R^2$, and RMSE for streamflow prediction is 0.70, 0.49, -12.5, 0.62, and 5.08 $m^3$ $s^{-1}$ by SWAT-SVR, and 0.36, -0.58, 88.39, 0.36, and 8.57 $m^3$ $s^{-1}$ by SWAT-CUP, respectively. It is clear that streamflow prediction from SWAT-CUP in the dry season had greater deviation in comparison with SWAT-SVR simulations. The developed model underestimated the dry season streamflow.

Streamflow prediction between the wet and dry seasons differed and wet season prediction easily obtained better performance. Low flows took place in dry seasons are a seasonal phenomenon, and their prediction is a challenging task in hydrology [58]. This difficulty may be attributed to the complexity of groundwater processes and the lack of effective evaluation criteria of low flows. Low flows in the dry season are typically generated from groundwater discharge or surface discharge from lakes, reservoirs, and marshes [58]. However, it is hard to investigate subsurface water discharge from nearby watersheds into a river channel in an unclosed watershed because of the limitation of hydrological measurement methods and the complexity of groundwater flow processes. Often these types of groundwater models are highly site-specific [59] or cover vast areas [60]. Furthermore, there are no effective and suitable statistical indicators to estimate the performance of low flows simulation. Both $R^2$ and NSE are known to put greater emphasis on high flows prediction and are sensitive to the hydrological regime, sample size or outliers [61]. Pushpalatha, Perrin [61] suggested using the objective function NSE of SqrtQ or lnQ for low flows evaluation.

The flow duration curves of observed versus simulated streamflow by SWAT-SVR are given in Fig 7 for each subwatershed. Fig 7 reveals that the developed model failed to capture extreme high flows with one exception (i.e. 07196090 in the dry season), but it worked well for various ranges of flow values especially for most medium flows and some low flows in the dry season. For example, simulations from 07195500, 07195430, 07196500, 07195865, and 07198000 in the wet season, and simulations from 07195800, 07195855, 07195865, 07195500, 07195430, 07196090, 07196500, 07197360, and 07198000 in the dry season matched observations well in medium and low flows. We noted that the flow duration curves of observations from 07196090 and 07196973 sites are steep. This is also because the length of flow data from the above two locations is 24 and 48 months, which only reflected the short and local temporal characters of flow duration.

## Model suitability analysis

To clearly reflect the spatial distribution of the SWAT-SVR model performance, we plotted the rating map of different models in validation during the wet and dry seasons (Fig 8). In the wet season, five models with ratings of "Good" came from 07195500, 07196090, 07196500, 07196500, and 07196900 sites where the flow discharge belonged to medium flows between 5

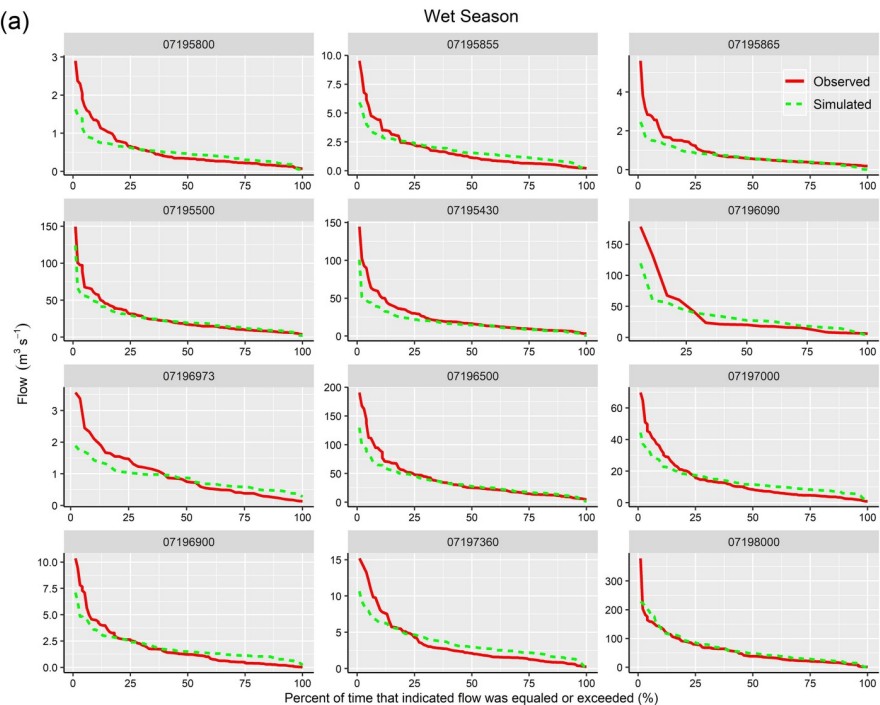

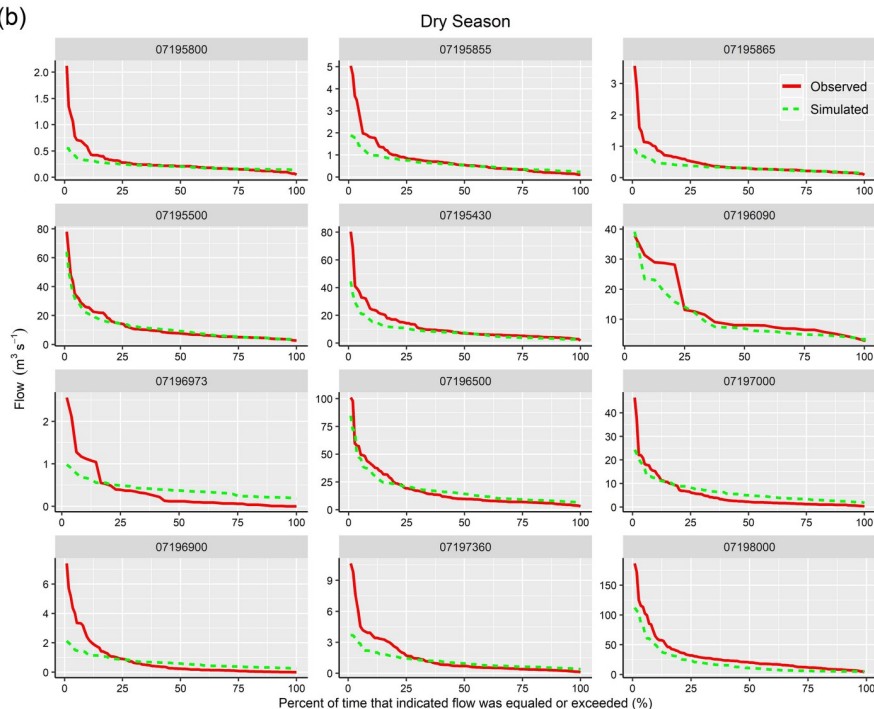

**Fig 7. Flow duration curves of observed versus simulated monthly streamflow during the wet and dry season with the SWAT-SVR model.**

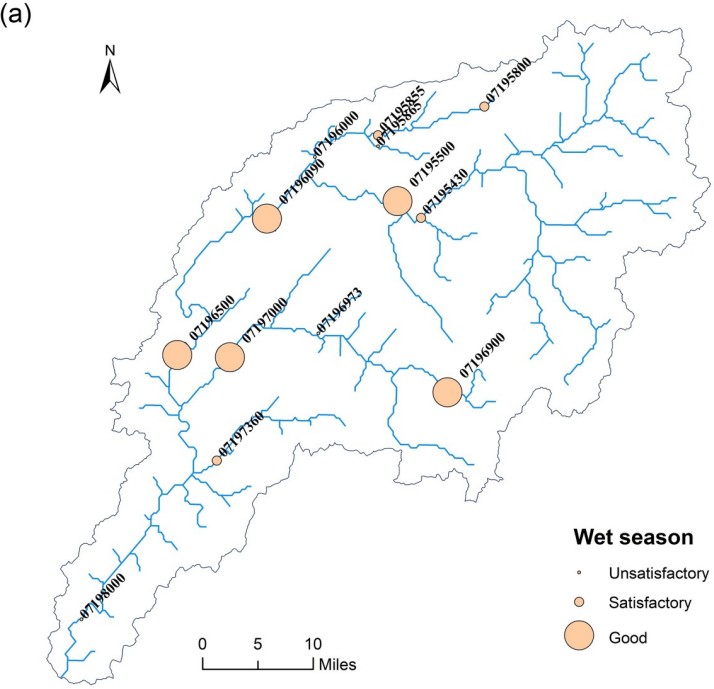

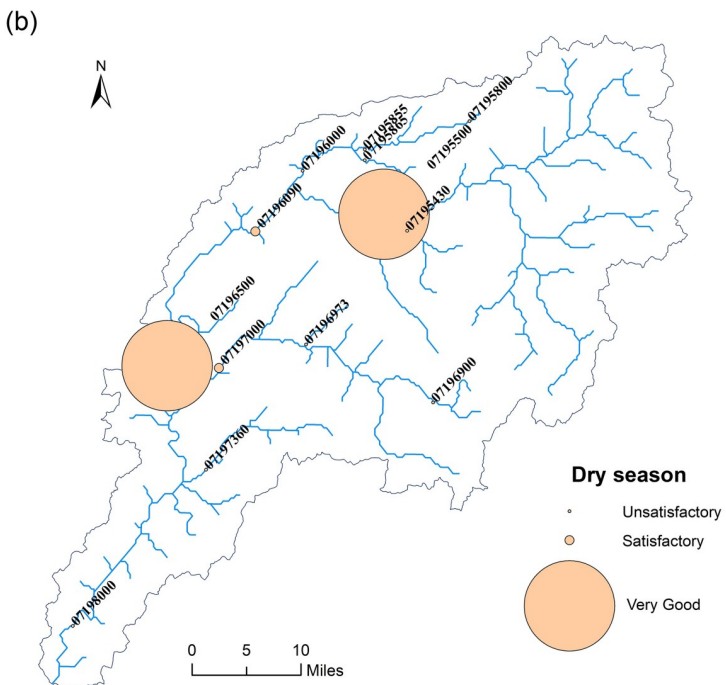

**Fig 8. SWAT-SVR model performance ratings during the (a) wet and (b) dry season.**

m³ s⁻¹ and 30 m³ s⁻¹ except 07196900 with low flows where flow discharge is less than 5 m³ s⁻¹. Four "Satisfactory" models came from 07195800, 07195855, 07197360, and 07195430 sites where the first three sites belonged to the low flows group except for 07195430 with medium flows. In the dry season, models from 07195500 and 07196500 sites had "Very Good" performance while the other two models from 07196090 and 07197000 sites were rated as "Satisfactory". All four of these models came from the medium flows group. Out of twelve models with medium flows, 07195500 and 07196500 had the best performance during the wet and dry seasons. The reason that SWAT-SVR cannot capture high flows is because events with a flow discharge larger than 30 m³ s⁻¹ were very rare (only account for 10.2% (263 data points) of total observations (2574 data points)). As a result, the number of high flows data in seasonal SVR calibration was less than 5.1% of total observations. Among high flows dataset, SVR cannot obtain enough training in calibration although the fact that SWAT generally overestimated these events (i.e. PBIAS is 21) is helpful to SVR training. The problem could be solved by adding more parameters controlling hydrological response such as precipitation, temperature and groundwater level to further train SVR. However, such an analysis is beyond the scope of our work. We also noted that the validation result from the outlet (07198000) of the IRW was unsatisfactory regardless of wet or dry season. This is because we know little about the operations of the upstream dam nearby 07198000 station, and this information has not been added into SWAT simulation. Meanwhile, this result also confirmed the opinion from Daggupati, Pai [25] that a single site calibration method (generally the outlet of a watershed) might not be suitable for simulations of a large watershed due to the spatial heterogeneity. In this case, the spatial calibration considering multiple sites is a more reliable method.

We plotted the relationship between estimating indicators and the upstream drainage area to further discover the spatial scale on which the model is applicable (Fig 9). In Fig 9, the y-axis is the value range of NSE, R², and RSR statistics; the x-axis represents the upstream drainage area of each station. The shaded region is 95% confidence interval of each indicator. We conducted the local polynomial regression analysis [62] on the above three indicators to find the trend of indicators change over the size of the watershed area. Fig 9a demonstrates that the

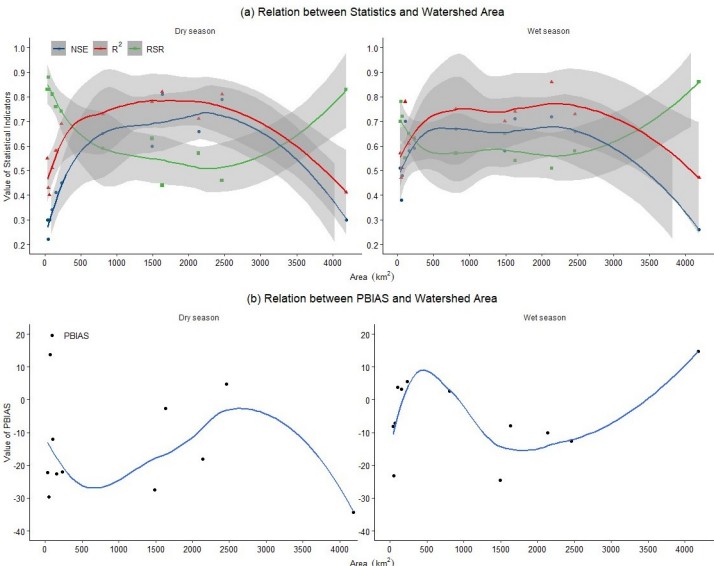

**Fig 9. The relation between estimation indicators and the upstream drainage area during the wet and dry season by validated SWAT-SVR.**

indicator of NSE, $R^2$, and RSR have similar changing patterns during the wet and dry seasons. The value of NSE and $R^2$ can stay at a high level, and RSR keeps a low value when the size of a watershed falls in the range of 500 to 3000 km$^2$. If these conditions from Fig 9a can be met; meanwhile, PBIAS value is small, and then the model would have better performance. The response of PBIAS value on the change of watershed size did not present a distinct pattern (Fig 9b). Therefore, we conclude that the developed SWAT-SVR model is applicable at sites with medium flows (i.e. 07195500, 07195430, 07196090, 07196500, and 07197000) where the upstream drainage area is between 500 and 3000 km$^2$.

### Streamflow prediction on yearly time series

To obtain an entire understanding of monthly streamflow prediction in the IRW, we combined the wet and dry seasons validated simulations, recalculated statistical indicators, and re-estimated overall model performance on the entire time series (i.e. calibration and validation periods are considered together). We summarized the overall performance indicators computed for SWAT-SVR and SWATCUP (Table 5). Table 5 shows 66.7% of twelve SWAT-SVR models had "Satisfactory" to "Very Good" performance ratings. The average RSR, NES, PBIAS, $R^2$, and RMSE is 0.62, 0.60, -8.34, 0.66, and 8.51 m$^3$ s$^{-1}$ for the developed model, respectively. The overall performance of twelve models on yearly time series is "Satisfactory". By comparison, only one site had a "Satisfactory" rating from SWAT-CUP. In most cases, the SWAT-SVR model outperformed the SWAT-CUP method.

We also plotted monthly streamflow hydrography for each site in Fig 10 to better explain where the developed model performed better than SWAT-CUP method. In Fig 10, all sites have similar hydrologic characteristics and they are all located in the IRW. The SWAT-SVR model works well for most medium flows and some low flows and can capture their timing and shape of rising and recession curves, but failed to capture extreme high flows on a monthly time scale (e.g. in the wet season of 2000, 2008, and 2011). We believe there are likely different drivers of hydrologic flow in wet and dry season that are not equivalently captured or modeled by SWAT, particularly because the purpose of SWAT development is not focused on flood prediction. As expected, the performance of SWAT-SVR heavily relied on the training data, it did not perform well when predicting high flows due to a small amount of training data in this study. However, we observed better prediction in the medium flow and few low flow

**Table 5. Overall performance ratings by SWAT-SVR and SWAT-CUP after combining wet and dry simulations.**

| Station | SWAT-SVR | | | | | | SWAT-CUP | | | | | |
|---|---|---|---|---|---|---|---|---|---|---|---|---|
| | RSR | NSE | PBIAS | $R^2$ | RMSE (m$^3$ s$^{-1}$) | Ratings | RSR | NSE | PBIAS | $R^2$ | RMSE (m$^3$ s$^{-1}$) | Ratings |
| 07195800 | 0.69 | 0.52 | -12.9 | 0.57 | 0.31 | Satisfactory | 0.71 | 0.50 | 14.8 | 0.55 | 0.31 | Unsatisfactory |
| 07195855 | 0.63 | 0.60 | -4.8 | 0.61 | 0.91 | Satisfactory | 0.88 | 0.22 | 46.4 | 0.49 | 1.26 | Unsatisfactory |
| 07195865 | 0.77 | 0.40 | -25.4 | 0.49 | 0.57 | Unsatisfactory | 0.83 | 0.31 | 29.5 | 0.44 | 0.61 | Unsatisfactory |
| 07195500 | 0.49 | 0.76 | -6.2 | 0.78 | 9.75 | Very Good | 0.64 | 0.58 | 35.2 | 0.70 | 12.89 | Unsatisfactory |
| 07195430 | 0.61 | 0.62 | -25.6 | 0.74 | 11.80 | Unsatisfactory | 0.57 | 0.68 | 20.6 | 0.71 | 10.91 | Satisfactory |
| 07196090 | 0.49 | 0.76 | -12.6 | 0.84 | 16.36 | Good | 0.46 | 0.78 | 32.2 | 0.84 | 15.49 | Unsatisfactory |
| 07196973 | 0.70 | 0.50 | -4.7 | 0.56 | 0.53 | Unsatisfactory | 0.96 | 0.07 | 60.6 | 0.45 | 0.72 | Unsatisfactory |
| 07196500 | 0.52 | 0.73 | -7.4 | 0.77 | 15.77 | Good | 0.66 | 0.56 | 37.1 | 0.68 | 19.92 | Unsatisfactory |
| 07197000 | 0.54 | 0.71 | 8.3 | 0.77 | 6.16 | Good | 0.78 | 0.40 | 61.0 | 0.65 | 8.87 | Unsatisfactory |
| 07196900 | 0.60 | 0.63 | -0.9 | 0.67 | 1.07 | Satisfactory | 0.96 | 0.07 | 90.2 | 0.58 | 1.69 | Unsatisfactory |
| 07197360 | 0.63 | 0.61 | -3.0 | 0.62 | 1.80 | Satisfactory | 0.87 | 0.24 | 62.9 | 0.57 | 2.51 | Unsatisfactory |
| 07198000 | 0.79 | 0.37 | -4.9 | 0.45 | 37.10 | Unsatisfactory | 0.93 | 0.13 | 49.5 | 0.42 | 43.57 | Unsatisfactory |

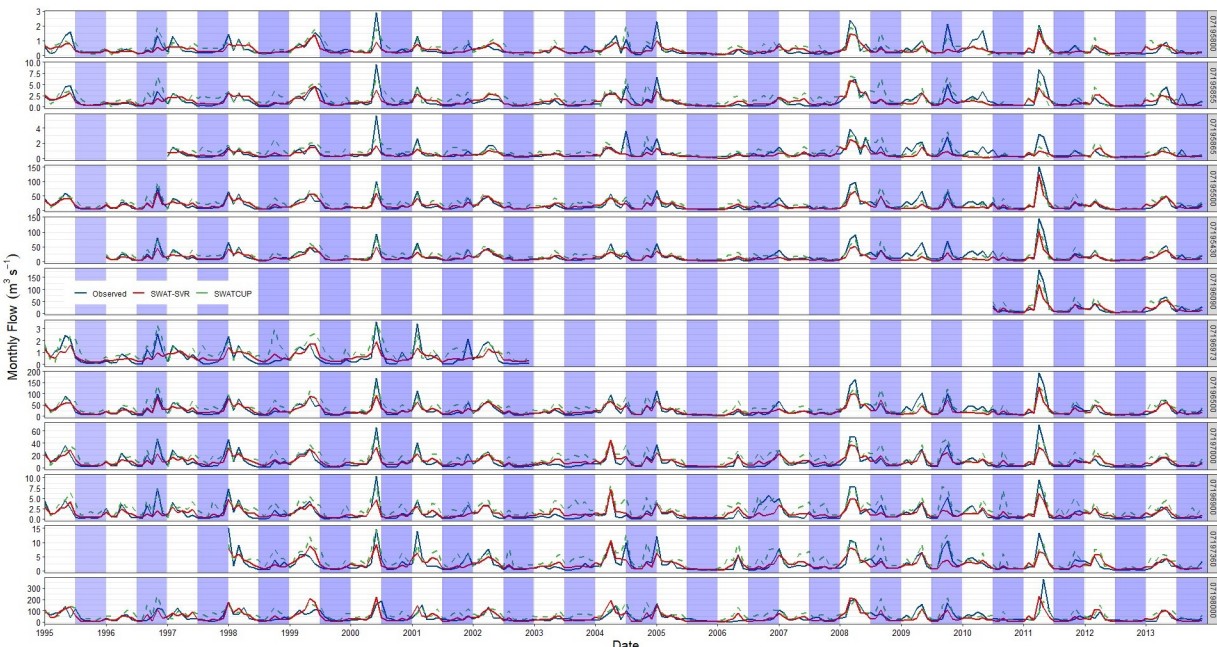

**Fig 10. Monthly streamflow time series from observation and validated simulation by SWAT-SVM and SWAT-CUP.** (The blue shade part represents the dry season from July to December).

conditions because SVR obtained enough training; another possible reason is that SVR captured a complicated nonlinear pattern from baseflow and groundwater patterns in the system that manifested in the high flow prediction.

Some of the dry seasons had more flow discharge than the wet seasons (e.g. the dry season of 1996, 2004, and 2009), and that was an error source of the SWAT-SVR model. The simulations from SWAT-CUP can capture extreme high flows (e.g. the wet season of 2000, 2008, and 2011), but far overestimated some medium flows and most low flows in the dry season (e.g. 1998, 2001, 2003, 2007, and 2010). There may be nonlinear drivers that exist due to other factors that are also difficult to incorporate during typical model calibration but better represented the system. Overall, the developed model can fit well with observations for most subwatersheds of the IRW.

In our study, the proposed method decreased the procedures of the SWAT model calibration and parameterization processing. Output streamflow from SWAT and the upstream drainage area were input into SVR where only three parameters needed to be verified. It made the parameter transfer of a hydrological model easier and feasible [63]. Additionally, we did not conduct the uncertainty analysis on the model but used strict criteria to estimate the model performance to limit the uncertainty of the SWAT-SVR model. Moreover, we used the spatial calibration and leave-one-out sampling method, meaning the validation work of any test watershed synthesized hydrologic information from the other 12 sub-watersheds. It is helpful for flow prediction at an ungauged or limited data watershed. In this sense, the developed model can serve as a regional tool as it integrates all information from nearby watersheds.

## Conclusions

This study developed a streamflow prediction model on a monthly time scale based on the SWAT model and the SVR method. Streamflow output from SWAT simulation and the

upstream drainage area were served as two input variables into SVR. The methodology considered various physical processes influencing flows change through integrating the SWAT model inside, as well as reducing time needed to calibrate and validate SWAT and time for feature selection in SVR while trying different parameter combinations. The overall performance of the model on the continuous time series is "Satisfactory" based on Table 5. The hybrid model predicted streamflow more accurately during the wet season than the dry season. Also, the model is likely applicable in situations that require better performance under medium flow conditions, for example, in this case, a watershed with medium flows with discharge between 5 m$^3$ s$^{-1}$ and 30 m$^3$ s$^{-1}$ where the upstream drainage area is between 500 to 3000 km$^2$. The strength of the proposed SVR approach is its capability to capture the intrinsic non-linear characteristics between rainfall-runoff while considering physical processes by integrating the SWAT model. Moreover, by using the spatial calibration and leave-one-out sampling method, the developed SWAT-SVR model can serve as a good regional tool for an ungauged or limited data watershed that has similar hydrologic characteristics with the IRW.

In cases where data are scarce, like an ungauged watershed, it is reasonable to apply proxy data and use machine learning techniques like SVM with physically based spatially distributed models, like SWAT, to produce high quality hydrologic prediction and, depending on the quantity of data available, describe more of the nonlinear variability that is often lost with conceptually built physical models that are inherently process weak [64]. Even though the calibration process may improve prediction without intrinsically including all physical processes [26], we believe this calibration approach can be incorporated into those process model predictions with a hybrid calibration procedure, like the one presented here. This approach may be a way to better represent the diversity of difficult to model hydrologic heterogeneity like groundwater discharge and nonlinearity that are contained within process model predictions often observed in physically based models within the constraints of current modeling practice, particularly in ungauged watersheds.

## Supporting information

**S1 Appendix.**
(DOCX)

## Acknowledgments

This research was performed while Dr. Lifeng Yuan held an NRC Research Associateship award at Robert S. Kerr Environmental Research Center, Ada, OK 74820. This work does not reflect the views of the US EPA, and no official endorsement should be inferred. We appreciate Dr. Yongping Yuan, Dr. Mohamed Hantush, Katherine Buckler, and Pat Bush for support of this paper. We are also thankful for Dr. Tibebe B. Tigabu and other two anonymous reviewers for their constructive comments.

## Author Contributions

**Conceptualization:** Lifeng Yuan, Kenneth J. Forshay.

**Data curation:** Lifeng Yuan.

**Formal analysis:** Lifeng Yuan.

**Funding acquisition:** Kenneth J. Forshay.

**Investigation:** Lifeng Yuan, Kenneth J. Forshay.

**Methodology:** Lifeng Yuan, Kenneth J. Forshay.

**Project administration:** Kenneth J. Forshay.

**Resources:** Kenneth J. Forshay.

**Supervision:** Kenneth J. Forshay.

**Validation:** Lifeng Yuan.

**Visualization:** Lifeng Yuan.

**Writing – original draft:** Lifeng Yuan, Kenneth J. Forshay.

**Writing – review & editing:** Lifeng Yuan, Kenneth J. Forshay.

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
