## [Decision Letter · Decision Letter 0]

25 Nov 2020

PONE-D-20-30823

Enhanced Streamflow Prediction with SWAT Using Support Vector Regression for Spatial Calibration: A Case Study in the Illinois River Watershed, U.S.

PLOS ONE

Dear Dr. Forshay,

Thank you for submitting your manuscript to PLOS ONE. After careful consideration, we feel that it has merit but does not fully meet PLOS ONE’s publication criteria as it currently stands. Therefore, we invite you to submit a revised version of the manuscript that addresses the points raised during the review process.

We look forward to receiving your revised manuscript.

Kind regards,

Mou Leong Tan

Academic Editor

PLOS ONE

Journal Requirements:

3. We note that Figures 1, 2 and 5 in your submission contain [ap/satellite images which may be copyrighted.

a. You may seek permission from the original copyright holder of Figures 1, 2 and 5 to publish the content specifically under the CC BY 4.0 license. 

4. Please include your tables as part of your main manuscript and remove the individual files. Please note that supplementary tables should remain as separate "supporting information" files.

Reviewers' comments:

Reviewer's Responses to Questions

**Comments to the Author**

1. Is the manuscript technically sound, and do the data support the conclusions?

Reviewer #1: Partly

Reviewer #2: Yes

Reviewer #3: Partly

2. Has the statistical analysis been performed appropriately and rigorously? 

Reviewer #1: Yes

Reviewer #2: Yes

Reviewer #3: I Don't Know

3. Have the authors made all data underlying the findings in their manuscript fully available?

Reviewer #1: No

Reviewer #2: Yes

Reviewer #3: Yes

4. Is the manuscript presented in an intelligible fashion and written in standard English?

Reviewer #1: Yes

Reviewer #2: Yes

Reviewer #3: Yes

5. Review Comments to the Author

Reviewer #1: This study proposed a method to improve monthly streamflow prediction performance by coupling a seasonal Support Vector Regression (SVR) model with the Soil and Water Assessment Tool (SWAT) model, and applied it in the Illinois River watershed (IRW), U.S. Overall, this paper presents an interesting approach for improving streamflow predictions. However, I think the following issues should be addressed before the paper is considered for publication.

1) I do not understand why the authors chose the approach to calibrate and validate the SWAT-VAR model by leaving out one station. This means that the authors need to develop 13 SWAT-SVR models, whose final parameter values could be rather different (unfortunately, the authors did not discuss this point in the paper). In this case, what should be the SWAT-VAR model for the entire watershed? In my opinion, the traditional approach that includes all stations but divides the study period into the calibration and validation periods works better here.

2) The SWAT model is a continuous simulation model. I could not fathom how the authors could run SWAT-CUP for dry and wet seasons independently. The authors have not provided any SWAT model parameter calibration results in the paper.

3) There are some logical flaws in the authors’ discussions related to Fig. 9. What is presented in Fig 9 is the evaluation statistics solely for the validated watershed. However, each SWAT-SVR model was developed using the data of the other 12 watersheds of various sizes. Performance at the single validated watershed is not sufficient to judge the model’s overall performance, let alone, to determine the application scope of the SWAT-SVR model. This judgement should be based on the model performance at all 13 watersheds. This is why I suggest the authors drop the “leaving-out-one-watershed” approach for calibration and validation.

4) The authors did not give any reason of including watershed area, but no other variable, in the SWAT-SVR model. Is it sufficient to include this single variable besides SWAT streamflow results in the model?

Reviewer #2: 1) The parameters considered in SWAT calibration and SWAT-SVR Calibration are not discussed. Is both of the calibration parameters chosen are the same for both model?

2) It is mentioned in the paper that SUFI 2 is being use for SWAT calibration, however for SWAT-SVR Calibration, how is it being conducted?

3) Author's use 5 statistical approaches to identify the model accuracy, however based on Table Table 3, only 3 statistical approaches rating has been shown, it will be better to include another 2 statistics.

4) Is is a bit unclear on how the SWAT-SVR being programmed, is it via Mathlab? The author may want to elaborate more on the system.

5) Figure 10 shows some high peak rainfall are unable to capture via both model, elaboration on this will hep future researcher to consider the factors.

6) Overall the paper is a good paper with some good analysis and explanation and may hep future researcher to conduct research on hydrological model.

Reviewer #3: The present work “Enhanced Streamflow Prediction with SWAT Using Support Vector Regression for Spatial Calibration: A Case Study in the Illinois River Watershed, U.S.” is interesting and original. Its main point of interest and originality is the development of a hybrid SWAT and Support Vector Regression (SVR) model based on 13 hydrologic gauging stations in Illinois River, US

However, there are some points that need clarification or re-consideration by the authors.

Introductions:

1. On page 4, line 68-69, the authors argue that several studies in the past have evaluated the performance of SWAT and SVM models in streamflow prediction separately, and the authors stated that few studies have coupled the two models. But the authors did not include those few studies and the drawbacks or gaps. Thus, the reviewer suggest to mention the past studies that focused on coupling of SWAT and SVM, and the novel idea of the current study.

Methodology:

2. On page 6, line 124 to 125, it was mentioned that multiple land use/soil/slope method was applied to define the HRUs in SWAT model with land use (10%), soil (10%) 125 and slope (5%) threshold. Is there any justification why these threshold values were selected?

3. In this manuscript, it seems that SWAT-CUP calibration approach was used and the modelled streamflow results are validated against measured ones. However, the authors did not mention the hydrologic parameters that control streamflow. When the authors discuss about the model performances, they compared SWAT-CUP against SWAT-SVR. But, it is difficult for the reader to understand easily how the model outcomes came especially for SWAT-CUP (example page 12, line 251 – 252). Moreover, the calibration and validation periods are not stated clearly,

4. One of the most important feature of SWAT-CUP is its capability to determine the uncertainty level of SWAT model prediction using sequential uncertainty fitting 2 (SUFI-2) algorithm. However, the current study used SWAT-CUP- (SUFI-2) as a tool of calibration and validation method, the level of model uncertainties was missed or not explained sufficiently why it was not included.

Result and discussions:

4. In the manuscript, it was mentioned that the predicted monthly streamflow by SWAT-SVR was more accurate during wet season that the dry season. Detail explanation is required why the model performances differ between the wet and dry seasons.

5. Page 12, line 251- 252, it was mentioned that “SWAT-CUP method also underestimated wet season streamflow but remarkably overestimated dry season streamflow. The SWAT-SVR model has approximately similar performances for the wet and dry seasons”. The reviewer believes that more discussions are required based on the feature of the two methods.

6. PLOS authors have the option to publish the peer review history of their article (what does this mean?). If published, this will include your full peer review and any attached files.

Reviewer #1: No

Reviewer #2: No

Reviewer #3: **Yes: **Tigabu, Tibebe B.

---

## [Author Response · Author response to Decision Letter 0]

2 Feb 2021

Response to editor and reviewer comments. Answer indicates the start of our response.

Editor Comments:

Journal Requirements:

Answer: We appreciate the Academic Editor’s concern on the format of our initial manuscript. We have formatted our manuscript according to above two template files. Please let us know if any aspect of our manuscript does not follow the format requirement of the PloS One journal. 

Answer: Yes, we have uploaded the study data in a stable, public repository as DOI: 10.23719/1520734 . Anyone can freely access our study data at doi.org/10.23719/1520734 upon publication. 

The upload data are divided into two groups: spatial data and time-series data. Spatial data are Digital elevation model (DEM) that obtained from Shuttle Radar Topography Mission (SRTM) 1 Arc-Second (about 30 m × 30 m) Global Database and downloaded from USGS website (https://earthexplorer.usgs.gov/, 01-28-2018) (Fig 2a). Land use and land cover (LULC) data was from the 2011 NLCD dataset (https://www.mrlc.gov/, 01-31-2018) (Fig 2b), and spatial resolution is 100 m × 100 m. Soil data came from the SSURGO database (https://websoilsurvey.nrcs.usda.gov/, 02-05-2018) (Fig 2c). Time-series data include climate and discharge data. Climate data obtained from the National Climatic Data Center (NCDC) (https://www.ncdc.noaa.gov/, 02-07-2018) (Fig 2d). Due to missing precipitation and temperature records from NCDC climate data from Jan. 1990 to Dec. 2013, we downloaded alternative Climate Forecast System Reanalysis (CFSR) data from the SWAT official website (https://globalweather.tamu.edu/, 01-31-2018), then filled missing NCDC data using climate data from the closest CFSR grid stations (not shown in Fig 2d). The streamflow data came from thirteen U.S. Geological Survey (USGS) hydrologic stations. These monthly discharge data can be accessed and downloaded by USGS official website (https://dashboard.waterdata.usgs.gov/app/nwd/?region=lower48).

3. We note that Figures 1, 2 and 5 in your submission contain [ap/satellite images which may be copyrighted.

a. You may seek permission from the original copyright holder of Figures 1, 2 and 5 to publish the content specifically under the CC BY 4.0 license. 

Answer: We have deleted the google map in Figure 1 within our manuscript to follow the requirement of publication. We used ESRI ArcGIS software to generate Figure 1, 2, and 8, and applied Microsoft Visio to generate Figure 5. And, EPA possesses the using licensed versions of ESRI ArcGIS and Microsoft Visio.

All data but the google map in the previous Figure 1 used in our research are derived from public data from public domains and produced by the authors. Anyone can download these data from the corresponding public repository listed. 

4. Please include your tables as part of your main manuscript and remove the individual files. Please note that supplementary tables should remain as separate "supporting information" files.

Answer: We have added the tables into the main body of our manuscript and removed the table document in the submitting system. 

Reviewers' comments:

Reviewer's Responses to Questions

Comments to the Author

1. Is the manuscript technically sound, and do the data support the conclusions?

Reviewer #1: Partly

Reviewer #2: Yes

Reviewer #3: Partly

2. Has the statistical analysis been performed appropriately and rigorously?

Reviewer #1: Yes

Reviewer #2: Yes

Reviewer #3: I Don't Know

3. Have the authors made all data underlying the findings in their manuscript fully available?

The PLOS Data policy requires authors to make all data underlying the findings described in their manuscript fully available without restriction, with rare exception (please refer to the Data Availability Statement in the manuscript PDF file). The data should be provided as part of the manuscript or its supporting information or deposited to a public repository. For example, in addition to summary statistics, the data points behind means, medians and variance measures should be available. If there are restrictions on publicly sharing data—e.g. participant privacy or use of data from a third party—those must be specified.

Reviewer #1: No

Reviewer #2: Yes

Reviewer #3: Yes

4. Is the manuscript presented in an intelligible fashion and written in standard English?

Reviewer #1: Yes

Reviewer #2: Yes

Reviewer #3: Yes

5. Review Comments to the Author

Reviewer #1: This study proposed a method to improve monthly streamflow prediction performance by coupling a seasonal Support Vector Regression (SVR) model with the Soil and Water Assessment Tool (SWAT) model, and applied it in the Illinois River watershed (IRW), U.S. Overall, this paper presents an interesting approach for improving streamflow predictions. However, I think the following issues should be addressed before the paper is considered for publication.

1) I do not understand why the authors chose the approach to calibrate and validate the SWAT-VAR model by leaving out one station. This means that the authors need to develop 13 SWAT-SVR models, whose final parameter values could be rather different (unfortunately, the authors did not discuss this point in the paper). In this case, what should be the SWAT-SVR model for the entire watershed? In my opinion, the traditional approach that includes all stations but divides the study period into the calibration and validation periods works better here.

Answer: In the revision on line 416 we have cited and described our approach. We appreciate reviewer’s concern on the technical detail of our manuscript. Here we used a spatial proxy approach. Generally, conventional calibration method of SWAT or other watershed models could be roughly divided into two groups: temporal and spatial calibrations. Temporal calibration is typically performed by splitting the available observed data into two datasets according to different periods: one for calibration, and another for validation. However, data may also be split spatially, with all available data at a given measured location assigned to the calibration phase and correspondingly perform the validation at one or more other gauges within the watershed. This method is spatial calibration (or also named spatial proxy basin approach). This spatial calibration approach is useful when users are faced with data-limited, ungauged situations or a study area is a large watershed. We present an approach and method that may be ultimately applied to an ungauged watershed. As Prasad Daggupati et al. (2015) pointed out, a regular temporal calibration method may not work well for a large size watershed due to potential spatial variability within the basin. Our study area is over 4,200 km2, and available data length from 13 USGS hydrologic station are not consistent (07196090 and 07196973 stations have only 42 and 96 data point in this study). Use of a temporal method to divide the datasets for calibration (say 70%) and validation subset (30%) due to a lack of similar numbers of wet, moderate, and dry years occurring in both periods may not be appropriate (Gan et al., 1997) albeit more familiar. 

Spatial calibration and validation approaches have been performed in several previous SWAT studies (e.g. Arnold et al.,2001, Van Liew and Garbrecht, 2003, Cao et al., 2006, Daggupati et al., 2015). Leave-one-out is a particular kind of spatial calibration method we used for this manuscript. For example, Navideh Noori et al. (2016) applied the leave-one-out method to develop a hybrid SWAT coupling ANN method. 

We did develop 26 SWAT-SVR models (for wet and dry seasons) and corresponding 26 SWAT models calibrated by SWAT-CUP. In 26 SWAT-SVR models, we used SWAT parameters by default (no calibration). For the SVR approach, 3 parameters (C, γ, ε) needed to be determined. The final value range of C for 26 SWAT-SVR models are from 53.0156 to 255.0156, the value of 𝛾 is 0.4, and ɛ is 0.00390625 for the wet season; the value of C falls between 32.0156 and 255.0156, 𝛾 is 1.2, and ɛ is 0.00390625 as well in the dry season. We can see C value have a wide range of change, but 𝛾 and ɛ are similar during the wet and dry seasons, respectively. This is because for any two runs there are always 11 common watersheds in SWAT-SVR. We made Fig. 5 (research flowchart) to help readers better understand our research processes. 

2) The SWAT model is a continuous simulation model. I could not fathom how the authors could run SWAT-CUP for dry and wet seasons independently. The authors have not provided any SWAT model parameter calibration results in the paper.

Answer: We have added an appendix e.g. Table 1 to help better describe our approach. We agree with reviewer’s opinion that SWAT model is a physically-based, continuous watershed model. However, that does not mean we cannot calibrate SWAT by dividing dry and wet seasons in SWAT-CUP. In fact, SWAT assumes that model parameters are season insensitive and attempt to identify ‘optimal’ values that would describe watershed behavior during dry and wet seasons. We considered and evaluated continuous unseparated calibration, but this assumption would compromise accuracy of model predictions. As we know, hydrologic models often perform poorly in simulation dry years in areas with large inter-annual variability in precipitation since the temporal variations in model parameters which exist in watersheds are not considered. Another reason is related with the objective function such as R2 and NSE, which can better reflect the hydrologic characteristics in wet periods than ones in dry periods. Misgana K. Muleta (2012), Dejian Zhang (2015), and Xin Gao(2018) published their paper which discussed in detail the method improvement of SWAT calibration by separating wet and dry seasons. 

Technically, it is straightforward to separate dry and wet season in SWAT-CUP because this software supports the leap-day, month, or year calibration. For example, the calibration file of separated wet and dry seasons in SWAT-CUP could be written as below:

No. Wet season Obs. No. Dry season Obs.

3 FLOW_OUT_3_1995 0.89 1 FLOW_OUT_1_1995 1.04

4 FLOW_OUT_4_1995 0.89 2 FLOW_OUT_2_1995 0.69

5 FLOW_OUT_5_1995 1.20 6 FLOW_OUT_6_1995 1.22

9 FLOW_OUT_9_1995 0.16 7 FLOW_OUT_7_1995 0.85

10 FLOW_OUT_10_1995 0.11 8 FLOW_OUT_8_1995 0.33

11 FLOW_OUT_11_1995 0.10 12 FLOW_OUT_12_1995 0.34

15 FLOW_OUT_3_1996 0.27 13 FLOW_OUT_1_1996 0.57

16 FLOW_OUT_4_1996 0.44 14 FLOW_OUT_2_1996 0.34

17 FLOW_OUT_5_1996 0.41 18 FLOW_OUT_6_1996 0.44

21 FLOW_OUT_9_1996 0.69 19 FLOW_OUT_7_1996 0.11

22 FLOW_OUT_10_1996 0.37 20 FLOW_OUT_8_1996 0.03

23 FLOW_OUT_11_1996 1.92 24 FLOW_OUT_12_1996 1.07

… … … … … …

Note that these observations must be ranked by time series order in SWAT-CUP to run successfully. 

In addition, we added the SWAT parameters initial range at SWAT-CUP calibration as below:

Table 1. The initial parameters and their range in calibration.

No Parameter Name1 Parameter Description Range Season

 If used in the wet season If used in the dry season

1 R__CN2.mgt SCS runoff curve number II -0.25-0.25 Yes Yes

2 V__ALPHA_BF.gw Baseflow alpha factor (1 day−1) 0–1 Yes Yes

3 V__GWQMN.gw Threshold depth of water in the shallow aquifer required for return flow to occur (mm H2O) 0–2000 Yes Yes

4 V__GW_REVAP.gw Groundwater “revap” coefficient 0.02–0.2 Yes Yes

5 V__EPCO.hru Plant uptake compensation factor 0–1 Yes Yes

6 R__SOL_K (1).sol Saturated hydraulic conductivity at the 1st soil layer (mm h−1) 30-102 Yes Yes

7 R__SOL_AWC (1).sol Available water capacity of the 1st soil layer (mm H2O mm soil−1) 0.08-0.2 Yes No

8 R__SOL_BD (1).sol Moist bulk density at the 1st soil layer (g cm−3) 1.3-1.45 Yes No

9 A__OV_N.hru Manning’s “n” value for overland flow 0.01–30 Yes No

10 A__CH_K2.rte Effective hydraulic conductivity in main channel alluvium (mm h−1) −0.01–500 Yes Yes

11 R__HRU_SLP.hru Average slope steepness (m m−1) 0–1 Yes Yes

12 V_RCHRG_DP.gw Deep aquifer percolation fraction 0-1 Yes Yes

13 A_CH_K1 Effective hydraulic conductivity in tributary channel alluvium 0-300 Yes No

14 V_ESCO.hru Soil evaporation compensation factor 0-1 No Yes

1 Note: “A__”, “V__” and “R__” mean an absolute increase, a replacement, and a relative change to the initial parameter values, respectively. 

3) There are some logical flaws in the authors’ discussions related to Fig. 9. What is presented in Fig 9 is the evaluation statistics solely for the validated watershed. However, each SWAT-SVR model was developed using the data of the other 12 watersheds of various sizes. Performance at the single validated watershed is not sufficient to judge the model’s overall performance, let alone, to determine the application scope of the SWAT-SVR model. This judgement should be based on the model performance at all 13 watersheds. This is why I suggest the authors drop the “leaving-out-one-watershed” approach for calibration and validation.

Answer: We developed 26 SWAT-SVR models for different seasons to represent the corresponding subwatersheds and verify the effectiveness of our hybrid method, not only for one or two models. We are thankful for the constructive comments from the reviewer and agree with the reviewer’s opinion that the single validated watershed is not enough to judge the model’s overall performance. This study attempts to improve the monthly streamflow at a large watershed by coupling SWAT and the SVR method. Spatial heterogeneity and temporal variability are intrinsically connected to watershed characteristics and in this system, we believe that segregating wet and dry season helps better describe the intrinsic dynamics of watershed hydrology, at least in this system. In this study, we developed 26 models for different subwatersheds. We don’t believe that a single continuous everywhere model is appropriate for such large basin given the seasonal variation and the likelihood that there are important changes to drivers of hydrology in the wet and dry seasons. 

4) The authors did not give any reason of including watershed area, but no other variable, in the SWAT-SVR model. Is it sufficient to include this single variable besides SWAT streamflow results in the model?

Answer: See lines 240-245. Our goal was to develop a model with few variable inputs .We appreciate reviewer’s concern on the model parameter selection of SWAT-SVR. We worked to develop an innovative method (a hybrid model) to improve the prediction of monthly streamflow in a large watershed. Since SVR is effectively a black-box model, we don’t consider all the physical parameters mechanistically that are interacting in potentially non-linear or chaotic ways inside the model, although SWAT does much of this, we wish to avoid more model variables or more complex mechanistic interactions that may not behave in a predictable way so that our SVR approach will better improve the model overall simulation. At first, we hoped to only use the streamflow from SWAT with default parameters as the sole input for SVR, but performance greatly improved when we added the variable -upstream drainage area into the simulations. We believe the upstream drainage area is the important parameter for this empirical mode because it plays a critical role at the early period of the hydrologic model, we describe the parameters on lines 233-239. Other variables such as terrain, soil, land use and land cover, weather, and others are included by SWAT. Here, we regarded the SWAT model essentially as a transfer function then hybridize it to improve prediction. 

Reviewer #2: 1) The parameters considered in SWAT calibration and SWAT-SVR Calibration are not discussed. Is both of the calibration parameters chosen are the same for both model?

Answer: We have added a table 1 in the appendix to better describe the model parameters. We agreed with reviewer’s opinion that the selection, sensitivity analysis and determination of SWAT parameters are very important for building a SWAT model. In our study, SWAT was applied with default values of parameters to simulate the streamflow at first, then we took the streamflow output from SWAT (no calibration) and upstream drainage area as input variables into the SVR model. For SVR, only three parameters need to be determined, and they are C, 𝛾, and ε which were determine by grid search and cross-validation. For SWAT-CUP, after parameter sensitivity analysis, we selected 13 parameters for wet season and 10 parameters for dry season, and applied the leave-one-out method and conducted spatial calibration for 13 stations. We attached initial parameters value range of SWAT-CUP before calibration for the wet and dry season. The ultimate parameter value or range of calibration for 13 stations are different one another because of the application of the spatial calibration method, so we don’t show all validation results from 13 stations in the manuscript. We focused on the model performance comparison of SWAT-SVR and SWAT-CUP.

Table 1. The initial parameters and their range in calibration.

No Parameter Name1 Parameter Description Range Season

 If used in the wet season If used in the dry season

1 R__CN2.mgt SCS runoff curve number II -0.25-0.25 Yes Yes

2 V__ALPHA_BF.gw Baseflow alpha factor (1 day−1) 0–1 Yes Yes

3 V__GWQMN.gw Threshold depth of water in the shallow aquifer required for return flow to occur (mm H2O) 0–2000 Yes Yes

4 V__GW_REVAP.gw Groundwater “revap” coefficient 0.02–0.2 Yes Yes

5 V__EPCO.hru Plant uptake compensation factor 0–1 Yes Yes

6 R__SOL_K (1).sol Saturated hydraulic conductivity at the 1st soil layer (mm h−1) 30-102 Yes Yes

7 R__SOL_AWC (1).sol Available water capacity of the 1st soil layer (mm H2O mm soil−1) 0.08-0.2 Yes No

8 R__SOL_BD (1).sol Moist bulk density at the 1st soil layer (g cm−3) 1.3-1.45 Yes No

9 A__OV_N.hru Manning’s “n” value for overland flow 0.01–30 Yes No

10 A__CH_K2.rte Effective hydraulic conductivity in main channel alluvium (mm h−1) −0.01–500 Yes Yes

11 R__HRU_SLP.hru Average slope steepness (m m−1) 0–1 Yes Yes

12 V_RCHRG_DP.gw Deep aquifer percolation fraction 0-1 Yes Yes

13 A_CH_K1 Effective hydraulic conductivity in tributary channel alluvium 0-300 Yes No

14 V_ESCO.hru Soil evaporation compensation factor 0-1 No Yes

1 Note: “A__”, “V__” and “R__” mean an absolute increase, a replacement, and a relative change to the initial parameter values, respectively. 

2) It is mentioned in the paper that SUFI 2 is being use for SWAT calibration, however for SWAT-SVR Calibration, how is it being conducted?

Answer: This is described on line 232-240. We first divided wet and dry season subsets for each station (total is 13). For dry or wet dataset, we applied leave-one-out method to determine the calibration and validation dataset. For each calibration dataset, we used grid search and k-fold cross-validation method to optimize the SVR parameters (C,γ, ε) by defining the upper and lower bound for each parameter and estimating the predicted accuracy of the model. In the k-fold cross-validation, the dataset was subdivided into k subsets of nearly equal size. In each step, the k-1 subsets were used to train the model while the remaining subset was used for validation. After found the optimized parameter set for each station, we can calculate the evaluation indicator for calibration, and applied this parameter set to validation dataset and obtained the statistics for the validation period. 

3) Author's use 5 statistical approaches to identify the model accuracy, however based on Table 3, only 3 statistical approaches rating has been shown, it will be better to include another 2 statistics.

Answer: We have added a citation to the table to represent that the table represents the more common statistical approaches used in SWAT calibration. In our study we indeed used 5 but adding the two parameters would be confusing for this table. We appreciate reviewer’s concern on how to evaluate the model performance. Table 3 was developed by Moriasi et al.,(2007) based on hundreds of studies. We believe the model performance will be easier to compare if we use this more common criterion to rate the model performance, but we also believe the added statistical approaches are also useful. However, we only followed the guideline developed by Moriasi et al., but didn’t develop further rating guideline for the additional statistics, which is beyond the scope of this paper. 

4) Is it a bit unclear on how the SWAT-SVR being programmed, is it via Mathlab? The author may want to elaborate more on the system.

Answer: The approach is described on line 263. We appreciate reviewer’s interest in the technical details of our programming. We didn’t use the Matlab platform. Instead, we used R and ‘e1071’ package to code the script for building the SVR model for different seasons at 13 stations. We used ‘tune’ function to conduct grid-search and cross-validate for the parameter set of C, γ, and ε, and used ‘svm’ function to build SVR model based on RBF kernel function. In this manuscript, we concentrate on the comparison of two different methods, results analysis and discussion parts. We don’t mind supplying the original R script of SWAT-SVR model development if that is necessary for publication. 

5) Figure 10 shows some high peak rainfall are unable to capture via both model, elaboration on this will help future researcher to consider the factors.

Answer: Thank you, we have appended the discussion into the revision on Figure 10 and lines 453-455. High peak streamflow cannot be captured well, partly reason might be that it is difficult to capture measured flow value during flooding by the flow sensor. And, the purpose of SWAT development is not for flood prediction, which has been explained at SWAT user manual. Also, the performance of SWAT-SVR heavily relied on the training data, it did not perform well when predicting high flows due to a small amount of training data in this study. 

6) Overall the paper is a good paper with some good analysis and explanation and may help future researcher to conduct research on hydrological model.

Answer: Thank you! We are thankful for reviewer’s comments. We also feel confident that our approach as well as the results and conclusions are useful. 

Reviewer #3: The present work “Enhanced Streamflow Prediction with SWAT Using Support Vector Regression for Spatial Calibration: A Case Study in the Illinois River Watershed, U.S.” is interesting and original. Its main point of interest and originality is the development of a hybrid SWAT and Support Vector Regression (SVR) model based on 13 hydrologic gauging stations in Illinois River, US

However, there are some points that need clarification or re-consideration by the authors.

Introductions:

1. On page 4, line 68-69, the authors argue that several studies in the past have evaluated the performance of SWAT and SVM models in streamflow prediction separately, and the authors stated that few studies have coupled the two models. But the authors did not include those few studies and the drawbacks or gaps. Thus, the reviewer suggests to mention the past studies that focused on coupling of SWAT and SVM, and the novel idea of the current study.

Answer: We have added some literature into the revision to summarize and elaborate the pervious related research and highlight the innovation of this study. The part of revision showed as follow:

“ Several works have evaluated the performance of SWAT and SVM in streamflow prediction [12, 19, 28]. Zhang, Srinivasan [12] et al. applied Artificial Neutral Network (ANN) and SVM methods to identify the optimal SWAT parameters to save the time cost of calibration and improve the efficiency of parameter calibration in two watersheds of the U.S. Jajarmizadeh, Kakaei Lafdani [28] et al. compared the monthly streamflow predictions from SWAT and SVM, and found the SVM model has a closer value for the average flow in comparison to the SWAT model. These efforts, however, either applied SVM in searching the optimal calibration parameters or built separate SWAT and SVR models, then estimated their running performance. Few studies have combined the two methods for a hybrid approach to streamflow prediction. Chiogna, Marcolini [19] et al. developed an SVM with SWAT model to predict hydropeaking in alpine watersheds in the Northeastern Italian. They used SVM to train the output of SWAT and found the SVM model can capture the fluctuation in streamflow. To the best of author’s knowledge, no study has coupled the SVM and SWAT and considered wet-dry change for streamflow prediction. The objective of this study is to show how a support vector regression (SVR) method to support SWAT calibration can be used to improve monthly streamflow prediction for different seasons in the IRW.”

Methodology:

2. On page 6, line 124 to 125, it was mentioned that multiple land use/soil/slope method was applied to define the HRUs in SWAT model with land use (10%), soil (10%) 125 and slope (5%) threshold. Is there any justification why these threshold values were selected?

Answer: We appreciate reviewer’s concern and awareness on how to determine threshold of land use, soil, and slope. We described the process of dividing threshold (land use (10%), soil (10%) 125 and slope (5%) threshold) is subjective and based on experience. Unfortunately, there is not presently a unified guideline on how to determine these thresholds for SWAT model development, but more accepted practice than explicit rules. The upstream drainage area we extracted is very close with the results published by USGS, so we believe our method are reasonable and the current threshold approach is appropriate. Defining and study of the impact of different thresholds dividing methods on simulations of SWAT is incredibly interesting, but beyond the scope of our research.

3. In this manuscript, it seems that SWAT-CUP calibration approach was used, and the modelled streamflow results are validated against measured ones. However, the authors did not mention the hydrologic parameters that control streamflow. When the authors discuss about the model performances, they compared SWAT-CUP against SWAT-SVR. But, it is difficult for the reader to understand easily how the model outcomes came especially for SWAT-CUP (example page 12, line 251 – 252). Moreover, the calibration and validation periods are not stated clearly,

Answer: We have appended the hydrologic parameters into the appendix for reader’s information. We appreciate the reviewer’s constructive comments. We need to emphasize that we used the spatial calibration method rather than temporal calibration method, and we applied the leave-one-out method to calibrate and validate the SWAT-SVR and SWAT-CUP. In each model run, the leave-one-out sampling method was applied to calibrate the SWAT-SVR model spatially. Out of n stations, one station was excluded for testing purposes, and the SWAT-SVR model was trained with the remaining (n-1) stations. This step was repeated until all stations had been removed once. SWAT-CUP passed through the same calibration process. We attached the calibration parameters and their range as below for reviewer’s reference.

Table 1. The initial parameters and their range in calibration.

No Parameter Name1 Parameter Description Range Season

 If used in the wet season If used in the dry season

1 R__CN2.mgt SCS runoff curve number II -0.25-0.25 Yes Yes

2 V__ALPHA_BF.gw Baseflow alpha factor (1 day−1) 0–1 Yes Yes

3 V__GWQMN.gw Threshold depth of water in the shallow aquifer required for return flow to occur (mm H2O) 0–2000 Yes Yes

4 V__GW_REVAP.gw Groundwater “revap” coefficient 0.02–0.2 Yes Yes

5 V__EPCO.hru Plant uptake compensation factor 0–1 Yes Yes

6 R__SOL_K (1).sol Saturated hydraulic conductivity at the 1st soil layer (mm h−1) 30-102 Yes Yes

7 R__SOL_AWC (1).sol Available water capacity of the 1st soil layer (mm H2O mm soil−1) 0.08-0.2 Yes No

8 R__SOL_BD (1).sol Moist bulk density at the 1st soil layer (g cm−3) 1.3-1.45 Yes No

9 A__OV_N.hru Manning’s “n” value for overland flow 0.01–30 Yes No

10 A__CH_K2.rte Effective hydraulic conductivity in main channel alluvium (mm h−1) −0.01–500 Yes Yes

11 R__HRU_SLP.hru Average slope steepness (m m−1) 0–1 Yes Yes

12 V_RCHRG_DP.gw Deep aquifer percolation fraction 0-1 Yes Yes

13 A_CH_K1 Effective hydraulic conductivity in tributary channel alluvium 0-300 Yes No

14 V_ESCO.hru Soil evaporation compensation factor 0-1 No Yes

1 Note: “A__”, “V__” and “R__” mean an absolute increase, a replacement, and a relative change to the initial parameter values, respectively. 

4. One of the most important features of SWAT-CUP is its capability to determine the uncertainty level of SWAT model prediction using sequential uncertainty fitting 2 (SUFI-2) algorithm. However, the current study used SWAT-CUP- (SUFI-2) as a tool of calibration and validation method, the level of model uncertainties was missed or not explained sufficiently why it was not included.

Answer: For SWAT-CUP, it is a relatively mature calibration software. It can easily and automatically obtain 95PPU. Hence, we can get the P-factor and r-factor directly. Discussing the uncertainty level of SWAT model prediction is direct for SWAT-CUP. However, SWAT-SVR presently is a relatively new R script, we haven’t consider how to calculate p-factor and r-factor within the script and do not focus on the uncertainty analysis and comparison of SWAT-SVR and SWAT-CUP output results, but that would be a good addition if our approach is adopted. We continue to improve the current scripts and develop more functions. We are considering ways to share our more innovative approaches in the future but this is probably beyond the scope of this manuscript. 

Result and discussions:

4. In the manuscript, it was mentioned that the predicted monthly streamflow by SWAT-SVR was more accurate during wet season than the dry season. Detail explanation is required why the model performances differ between the wet and dry seasons.

Answer: We have added line 366:

 “Low flows took place in dry seasons are a seasonal phenomenon, and their prediction is a challenging task in hydrology [58]. This difficulty may be attributed to the complexity of groundwater processes and the lack of effective evaluation criteria of low flows. Low flows in the dry season are typically generated from groundwater discharge or surface discharge from lakes, reservoirs, and marshes [58]. However, it is hard to investigate subsurface water discharge from nearby watersheds into a river channel in an unclosed watershed because of the limitation of hydrological measurement methods and the complexity of groundwater flow processes. Often these types of groundwater models are highly site-specific [59] or cover vast areas [60]. Furthermore, there are no effective and suitable statistical indicators to estimate the performance of low flows simulation. Both R2 and NSE are known to put greater emphasis on high flows prediction and are sensitive to the hydrological regime, sample size or outliers [61]. Pushpalatha, Perrin [61] suggested using the objective function NSE of SqrtQ or lnQ for low flows evaluation.”

5. Page 12, line 251- 252, it was mentioned that “SWAT-CUP method also underestimated wet season streamflow but remarkably overestimated dry season streamflow. The SWAT-SVR model has approximately similar performances for the wet and dry seasons”. The reviewer believes that more discussions are required based on the feature of the two methods.

Answer: We have added extra analysis and explanation into the revision. 

“Based on the value of PBIAS, the SWAT-SVR model slightly underestimated monthly streamflow for each watershed during the wet and dry seasons, and SWAT-CUP method also underestimated wet season streamflow but remarkably overestimated dry season streamflow. The average of NSE and R2 of 13 sites decreased from 0.92 and 0.92 in the wet season to -0.16 and 0.55 in the dry season, respectively. This results are consistent with Zhang, Chen [43]’s study in which the SWAT model can produce good simulations for the wet season but poor simulations for the dry season. The possible reason is that R2 and NSE are sensitive to extremely large number (i.e. high flows took place in the wet season). Compared with the performance of SWAT-CUP, the SWAT-SVR model has approximately similar performances for the wet and dry seasons. Although the simulation results from SWAT-CUP had better overall performance (i.e. 53.8% of the runs had “Very Good” ratings) than those of the SWAT-SVR model in the wet season, SWAT-CUP failed to estimate monthly streamflow in the dry season, in which all runs were identified as “Unsatisfactory” ratings. We also noted that the variation of statistics is small, and the value of each indicator is close between different SWAT-SVR models. This is because a single SWAT-SVR watershed model was built based on eleven other common watersheds in calibration.”

6. PLOS authors have the option to publish the peer review history of their article (what does this mean?). If published, this will include your full peer review and any attached files.

Do you want your identity to be public for this peer review? For information about this choice, including consent withdrawal, please see our Privacy Policy.

Reviewer #1: No

Reviewer #2: No

Reviewer #3: Yes: Tigabu, Tibebe B.

Answer: Thank you to each reviewer. We respect and appreciate they time and effort to review our work.

Ken Forshay and Lifeng Yuan.

---

## [Decision Letter · Decision Letter 1]

1 Mar 2021

Enhanced Streamflow Prediction with SWAT Using Support Vector Regression for Spatial Calibration: A Case Study in the Illinois River Watershed, U.S.

PONE-D-20-30823R1

Dear Dr. Forshay,

We’re pleased to inform you that your manuscript has been judged scientifically suitable for publication and will be formally accepted for publication once it meets all outstanding technical requirements.

Kind regards,

Mou Leong Tan

Academic Editor

PLOS ONE

Additional Editor Comments (optional):

I think that the authors did a great job addressing most of the comments. Well done.

Reviewers' comments:

Reviewer's Responses to Questions

**Comments to the Author**

1. If the authors have adequately addressed your comments raised in a previous round of review and you feel that this manuscript is now acceptable for publication, you may indicate that here to bypass the “Comments to the Author” section, enter your conflict of interest statement in the “Confidential to Editor” section, and submit your "Accept" recommendation.

Reviewer #1: All comments have been addressed

Reviewer #2: All comments have been addressed

Reviewer #3: All comments have been addressed

2. Is the manuscript technically sound, and do the data support the conclusions?

Reviewer #1: Partly

Reviewer #2: Yes

Reviewer #3: Yes

3. Has the statistical analysis been performed appropriately and rigorously? 

Reviewer #1: Yes

Reviewer #2: Yes

Reviewer #3: Yes

4. Have the authors made all data underlying the findings in their manuscript fully available?

Reviewer #1: Yes

Reviewer #2: Yes

Reviewer #3: Yes

5. Is the manuscript presented in an intelligible fashion and written in standard English?

Reviewer #1: Yes

Reviewer #2: Yes

Reviewer #3: Yes

6. Review Comments to the Author

Reviewer #1: I do not quite agree with the authors' statement that the studied watershed (<5000 km2) is a large watershed that requires spatial calibration. Meanwhile, the improvement of prediction is more useful if applied on daily streamflows instead of monthly streamflows. However, since the journal does not require innovation, I think it is better for me to leave the decision to the editor and the other two reviewers.

Reviewer #2: The authors answers all the concerns addressed and justified in written comments on one issue and I am satisfied with the justification. Overall I believe the paper is sound and rigid and ready for publication.

Reviewer #3: The authors have done a great job in responding to all the comments given for the first submission. Based on their the additional efforts and responses, now it seems that manuscript addressed all my comments adequately.

7. PLOS authors have the option to publish the peer review history of their article (what does this mean?). If published, this will include your full peer review and any attached files.

Reviewer #1: No

Reviewer #2: **Yes: **Nor Faiza Abd Rahman

Reviewer #3: No

---

## [Editor Report · Acceptance letter]

29 Mar 2021

PONE-D-20-30823R1 

Enhanced Streamflow Prediction with SWAT Using Support Vector Regression for Spatial Calibration: A Case Study in the Illinois River Watershed, U.S. 

Dear Dr. Forshay:

I'm pleased to inform you that your manuscript has been deemed suitable for publication in PLOS ONE. Congratulations! Your manuscript is now with our production department. 

Kind regards, 

on behalf of

Dr. Mou Leong Tan 

Academic Editor

PLOS ONE